# HCPD-CA High-resolution climate projection dataset in Central Asia

Yuan Qiu[1], Jinming Feng[1], Zhongwei Yan[1], and Jun Wang[1]

1 Key Laboratory of Regional Climate-Environment for Temperate East Asia (RCE-TEA), Institute of Atmospheric Physics, Chinese Academy of Sciences, Beijing 100029, China

Correspondence to: Jinming Feng, fengjm@tea.ac.cn

## Abstract

Central Asia (referred to as CA) is one of the climate change Hot-Spots due to the fragile ecosystems, frequent natural hazards, strained water resources, and accelerated glacier melting, which underscores the need of high-resolution climate projection datasets for application to vulnerability, impacts, and adaption assessments in this region. In this study, a high-resolution (9km) climate projection dataset over CA (the HCPD-CA dataset) is derived from dynamically downscaled results based on multiple bias-corrected global climate models and contains four geostatic variables and ten meteorological elements that are widely used to drive ecological and hydrological models. The reference and future periods are 1986-2005 and 2031-2050, respectively. The carbon emission scenario is Representative Concentration Pathway (RCP) 4.5. The evaluation shows that the data product has good quality in describing the climatology of all the elements in CA despite some systematic biases, which ensures the suitability of the dataset for future research. Main features of projected climate changes over CA in the near-term future are strong warming (annual mean temperature increasing by 1.62-2.02°C) and significant increase in downward shortwave and longwave flux at surface, with minor changes in other elements (e.g., precipitation, relative humidity at 2m, and wind speed at 10m). The HCPD-CA dataset presented here serves as a scientific basis for assessing the potential impacts of projected climate changes over CA on many sectors, especially on ecological and hydrological systems. It has the DOI https://doi.org/10.11888/Meteoro.tpdc.271759 (Qiu, 2021).

## 1. Introduction

Central Asia (referred to as CA, Fig. 1a) has complex terrain and diverse climates and is among the most vulnerable regions to climate change due to fragile ecosystems (Zhang et al., 2016;Seddon et al., 2016;Gessner et al., 2013), frequent natural hazards (Thurman, 2011;Burunciuc, 2020), strained water resources (Frenken, 2013), and accelerated glacier melting (Narama et al., 2010;Sorg et al., 2012), which underscores the need to achieve high-resolution climate projection datasets for application to vulnerability, impacts, and adaption assessments. Global climate models (GCMs) can describe the response of the global circulation to large-scale

forcing, such as greenhouse gases and solar radiation (Giorgi, 2019). But their horizontal resolutions are too coarse to account for the effects of local-scale forcing and processes, such as complex topography, land cover distribution, and dynamical processes occurring at the mesoscale (Giorgi et al., 2016;Qiu et al., 2017;Torma et al., 2015). To obtain the accurate information on region-scale climate change, dynamical downscaling has been developed and widely applied in regional climate projections over many areas, like East Asia (Zou and Zhou, 2016;Tang et al., 2016;Jung et al., 2015;Jiang et al., 2021;Ji and Kang, 2013;Hong et al., 2017;Guo et al., 2021;Bao et al., 2015;Zou and Zhou, 2017), North America (Wang and Kotamarthi, 2015;Racherla et al., 2012;Pierce et al., 2013;Giorgi et al., 1994;Di Luca et al., 2013, 2012;Wang et al., 2015), and Europe (Vautard et al., 2013;Jacob et al., 2014;Kotlarski et al., 2014;Fischer et al., 2015;Kotlarski et al., 2015;Torma et al., 2015;Giorgi et al., 2016;Zittis et al., 2019;Jacob et al., 2020;Déqué et al., 2007;Gao et al., 2006;Im et al., 2010). Some efforts have also been devoted on regional climate projection in CA with the dynamical downscaling method (Zhu et al., 2020;Ozturk et al., 2017;Mannig et al., 2013). However, their resolutions are still low (≥30km), especially for the mountainous areas in the southeast. Moreover, most of the previous RCM simulations in CA used a single GCM as the lateral boundary conditions, which harbor high uncertainties in the projected climate changes.

The present authors carried out a study that involves the dynamical downscaling of multiple bias-corrected GCMs for the CA region with an unprecedented horizontal resolution of 9km. The future simulation period is set as 2031-2050 under Representative Concentration Pathway (RCP) 4.5, with the reference period of 1986-2005. The simulated surface air temperature and precipitation have been evaluated in a recent study (Qiu et al., 2021) and meanwhile basic features of the projected climate changes have been demonstrated. The results show that the high-resolution RCM simulations can well capture the local temperature and precipitation in CA and detect significant warming, severer heatwaves, and drier conditions in this region in the near-term future.

To satisfy the urgent need of high-resolution climate data for assessing the potential impacts of the projected climate changes over CA on many sectors, especially on ecological and hydrological systems, the HCPD-CA (High-resolution Climate Projection Dataset in CA) dataset is derived from the 9-km-resolution downscaled results, which includes four geostatic (time-invariant) variables and ten meteorological elements (Table 1) that are widely used to drive ecological and hydrological models. The geostatic variables are terrain height (HGT, m), land use category (LU_INDEX, 21 categories), land mask (LANDMASK, 1 for land and 0 for water), and soil category (ISLTYP, 16 categories). The meteorological elements are daily precipitation (PREC, mm/day), daily mean/maximum/minimum temperature at 2m (T2MEAN/T2MAX/T2MIN, K), daily mean relative humidity at 2m (RH2MEAN, %), daily mean eastward and northward wind at 10m

(U10MEAN/V10MEAN, m/s), daily mean downward shortwave/longwave flux at surface (SWD/LWD, W/m$^2$), and daily mean surface pressure (PSFC, Pa). The present paper is to introduce this dataset to the community. Sect. 2 describes the regional model and experiments. Model evaluation and projected changes in the meteorological elements are in Sect. 3. Added values of using 9-km resolution respect to using coarser resolutions are discussed in Sect. 4 as well as uncertainties of the evaluation and the HCPD-CA dataset. Sect. 5 describes access to the data product and all codes and tools. Main results are concluded in Sect. 6.

## 2 Model and experiments

### 2.1 Regional model

The Weather Research and Forecasting (WRF) model with version 3.8.1 (Skamarock et al., 2008) is used to downscale the GCMs. It has two domains (Fig. 1b). The outer one covers a large region, with a 27-km resolution and 290×205 grids. The inner one covers the CA region, with a 9-km resolution and 409×295 grids. The model has 33 levels in the vertical direction with its top fixed at 50 hPa. Its physical schemes are set based on our previous work about the sensitivity analysis of physical parameterizations in the WRF model for local climate simulations in CA (Wang et al., 2020). Details about the optimal physical schemes are in Qiu et al. (2021). Spectral nudging with a weak coefficient of $3 \times 10^{-5}$ is applied in the outer domain (not in the inner one), which prevents possible model drift during the long-term integration by relaxing the model simulations of wind, temperature, and moisture toward the driving conditions. In addition to greenhouse gases and solar constant, the WRF model also considers other external forcing, such as aerosols, volcanoes, and ozone, to make its inner external forcing consistent with the driving GCMs.

The geogrid program in the WRF model is to define the simulation domains, and interpolate various terrestrial datasets to the model grids (Wang et al., 2007). First, geogrid computes the latitude, longitude, and map scale factors at every grid point. Then, it interpolates terrain height, land use category, soil category and other time-invariant data to the model grides. Global datasets of each of these fields are provided through the WRF download page (https://www2.mmm.ucar.edu/wrf/users/download/get_sources_wps_geog.html). The HCPD-CA dataset contains four of the geostatic variables. In them, the terrain height (HGT) data (Fig. S1) is from the United States Geological Survey (USGS) GTOPO30 elevation dataset, the land use category (LU_INDEX) data (Table S1 and Fig. S2) is from the Moderate Resolution Imaging Spectroradiometer (MODIS) 21 category land dataset, the soil category (ISLTYP) data (Table S2 and Fig. S3) is from the global 5-minute United Nation FAO soil category dataset, and the land mask (LANDMASK) data (Fig. S4) is calculated based on LU_INDEX with the condition that the value of a grid cell is set as 1 (0) if land (water)

area at least accounts for 50%.

## 2.2 Bias-correction technique

MPI-ESM-MR (referred to as MPI, Table 2), CCSM4 (CCSM), and HadGEM2-ES (Had) from Phase 5 of the Coupled Model Intercomparison Project (CMIP5) are selected to drive the regional model. The reasons why we chose these three GCMs are as follows: they can provide all the variables that are needed to drive the regional model; they have relatively high horizontal resolutions (Table 2) among the CMIP5 models; they have fairly good performance in simulating the local temperature and precipitation in CA (see Fig. S1 and S3 in Qiu et al., 2021), though systematic biases exist partially due to their coarse resolutions. Since all GCMs suffer from some forms of bias (Done et al., 2015;Ehret et al., 2012;Liang et al., 2008;Xu and Yang, 2012) that may propagate down to the RCM outputs, the bias-correction technique developed by Bruyère et al. (2014) is applied in this study to correct the climatology of the GCMs and meanwhile allow synoptic and climate variability to change.

Six-hourly GCM data in a 25-year base/future period (1981-2005/2026-2050), hereafter referred to as $GCM_{BP}/GCM_{FP}$, are broken down into the 25-year mean 6-hourly annual cycle over the base period ($\overline{GCM_{BP}}$) plus a 6-hourly perturbation term ($GCM_{BP}'/GCM_{FP}'$):

$$GCM_{BP} = \overline{GCM_{BP}} + GCM_{BP}' \tag{1}$$

$$GCM_{FP} = \overline{GCM_{BP}} + GCM_{FP}' \tag{2}$$

The ERA-Interim reanalysis data (Dee et al., 2011, Table 2) as "observations" ($Obs$) is similarly broken down into the mean annual cycle ($\overline{Obs}$) and a perturbation term ($Obs'$):

$$Obs = \overline{Obs} + Obs' \tag{3}$$

The bias corrected GCM data for the base/future period, $GCM_{BP}{}^*/GCM_{FP}{}^*$, is then constructed by replacing $\overline{GCM_{BP}}$ from Eq. 1/2 with $\overline{Obs}$ from Eq. 3:

$$GCM_{BP}{}^* = \overline{Obs} + GCM_{BP}' \tag{4}$$

$$GCM_{FP}{}^* = \overline{Obs} + GCM_{FP}' \tag{5}$$

Eq. 1-5 are applied to all the variables required to generate the initial and lateral boundary conditions for the WRF model: zonal and meridional wind, geopotential height, air temperature, relative humidity, sea surface temperature, mean sea level pressure, etc. In a recent study (Qiu et al., 2021), we conducted the sensitivity analysis of using the bias-correction technique, to quantify its contribution to improving the RCM simulation. The results show that using the bias-correction technique largely reduced the biases in the simulated annual and seasonal precipitation over CA relative to not using it and slightly improved the model's skill in simulating the spatial pattern of precipitation (see Fig. 4 in Qiu et al., 2021).

The bias-corrected CCSM4 outputs (DOI: https://doi.org/10.5065/D6DJ5CN4) is produced by Bruyère et al. (2014) with a 25-year base period (1981-2005) during the bias correction. In this study, we produced the bias-corrected MPI-ESM-MR and HadGEM2-ES outputs with the same base period as them. Note that the base period used during the bias correction is not necessary to be consistent with the reference period (1986-2005) of the RCM simulations.

## 2.3 Experiments

The RCM simulations with the bias-corrected GCMs (MPI, CCSM, and Had) as the driving data are referred to as WRF_MPI_COR, WRF_CCSM_COR, and WRF_Had_COR, respectively ("COR" means using the bias-correction technique). The reference-period simulations are from December 1, 1985 to December 31, 2005 and the future runs are from December 1,2030 to the end of 2050 under a moderate carbon emission scenario RCP 4.5, which is arguably the most policy-relevant scenario as the Nationally Determined Contributions (NDCs) greenhouse gas emissions framework would produce similar temperatures trajectories (Gabriel and Kimon, 2015). The first month in each simulation is discarded as spin up. Fig. 2 shows the flow chart to produce the HCPD-CA dataset. The procedure can be divided into four steps. First, a sensitivity analysis of physical parameterizations in the WRF model was done and then we identified the optimal physical parameterizations combination for WRF for regional climate studies over CA. Second, the original GCMs are bias corrected and the bias-corrected GCMs are used to drive the WRF model with the optimal physical schemes. Third, we conducted the dynamical downscaling over CA and produced 9-km resolution downscaled results. At last, the HCPD-CA dataset with certain variables and standard file formats is derived from the downscaled results.

# 3 Results

## 3.1 Model evaluation

In Qiu et al. (2021), the key meteorological elements, surface air temperature and precipitation in the RCM simulations, have been evaluated with both gridded observations and stations' data (see Sect. 3.1 in the paper) and the results show good skills of the regional model in simulating the local temperature and precipitation in CA during the reference period (1986-2005). Accordingly, the ten meteorological elements (including surface air temperature and precipitation) in the HCPD-CA dataset are evaluated here, to show the validity and applicability of the dataset. Note that daily mean wind speed at 10m (referred to as WS10MEAN) instead of U10MEAN and V10MEAN is evaluated.

Version 4 of the Climatic Research Units gridded Times Series (CRU TS v4, Harris et al., 2020, Table 2) is applied to evaluate T2MEAN/T2MAX/T2MIN and the land component of the fifth generation of European reanalysis (ERA5-Land, Hersbach et al., 2020, Table 2) is used as "observations" to evaluate other elements. Before the evaluation, the RCM outputs are interpolated to the grides of CRU TS v4 (ERA5-Land) with the distance-weighted average (bilinear) method. We found that both on the annual and seasonal scales, the interpolation methods conserved the area averaged values in the model outputs with a bias of less than 1-2% between the original and new grids. We thus concluded that our choice of interpolation procedure does not affect the main conclusions of our work.

The high-resolution downscaled results (WRF_MPI_COR, WRF_CCSM_COR, and WRF_Had_COR) are very close to the observational data in simulating the climatology of all the elements in CA on both annual and seasonal scales (Fig. 3-5, seasonal results not shown). For instance, the spatial correlation coefficients (SCCs) of all the annual mean values (except WS10MEAN) over CA are larger than 0.80. The SCCs of annual mean WS10MEAN over CA are relatively small, in a range of 0.54-0.64. The simulated annual mean T2MEAN over the very north of Kazakhstan and the Pamirs has cold bias and that over other areas generally has warm bias (Fig. S5a-c). However, the bias over most of CA is within -2~2℃. The annual mean RH2MEAN is generally underestimated over CA except some areas in the northern part and the Aral Sea (Fig. S6a-c). The RCM simulations commonly overestimate the annual mean WS10MEAN over the mountainous areas (Fig. S6d-f). Stronger annual mean SWD prevails in CA in each simulation (Fig. S7a-c), with the mean errors (MEs) over the whole region in a range of 27.72-31.43 W/m$^2$. Meanwhile, the regional model slightly underestimates annual mean LWD (Fig. S7d-f). The bias in annual mean PSFC is very small over the majority of CA (Fig. S7g-i). Table S3 summarizes the statistic metrics [SCCs, RMSEs, and mean errors (MEs)] of all the annual mean variables over both CA and its climate subregions [northern CA (NCA), middle CA (MCA), southern CA (SCA), and the mountainous areas (MT), see their scopes in Fig. 1c], to help the readers easily check the quality of this data product in the areas they are interested.

Fig. 6 shows mean annual cycle of the monthly values averaged over CA. It is seen that the model outputs are generally close to the observations. The warm bias in T2MEAN mainly occurs during May-August (Fig. 6a). The overestimation of SWD occurs throughout the year, with the bias larger in the warm seasons than in the cold seasons (Fig. 6e). The results of T2MAX and T2MIN are similar to those of T2MEAN (not shown here).

To sum up, the evaluation shows that the HCPD-CA dataset has good quality in describing the climatology of all the meteorological elements in CA despite some systematic biases (e.g., stronger SWD), which ensures the suitability of the dataset for assessment of future risks from climate change in CA.

## 3.2 Projected climate changes

Fig. 7 shows projected changes of the annual mean values in CA during 2031-2050, relative to 1986-2005. All the RCM simulations exhibit significant warming over CA in the near-term future, with the annual mean T2MEAN increasing by 1.62-2.02°C (Fig. 7a-c, range depending on the simulation). Pronounced warming is found in the north, which is attributed to the snow and surface albedo feedback (Qiu et al., 2021). Interestingly, enhanced warming projected in many mountainous regions around the world (Palazzi et al., 2019;Pepin et al., 2015;Rangwala et al., 2013) is not found in CA (also see Fig. 7-8 in Qiu et al. (2021)). It poses a question if the responses of ecological and hydrological systems to future warming in the Tien Shan and Pamirs differ from those in other mountains, like Tibetan Plateau/Himalayas and Alps.

The annual mean precipitation (PREC) is projected to sightly increase by 0.01-0.02 mm/day (Fig. 7d-f). However, changes in few areas passed the significance test. The annual mean RH2MEAN is simulated to sightly decrease by 0.68-1.28% (Fig. 7g-i), which suggests a drier condition in CA in the coming decades and may affect the physical and chemical properties of the local vegetations. Changes in wind speed (WS10MEAN) are inconsistent among the RCM simulations (Fig. 7j-l). WRF_MPI_COR shows a slight increase of 0.02m/s while others show a slight decrease, which highlights the uncertainties in the projected changes. Downward shortwave/longwave flux (SWD/LWD) are projected to significantly increase by 3.47-4.28 W/m$^2$ (Fig. 7m-o) and 7.13-9.61 W/m$^2$ (Fig. 7p-r), respectively. Surface pressure (PSFC) is simulated to slightly increase by 0.15-0.70 hPa in CA (Fig. 7s-u).

To sum up, main features of projected climate changes in CA in the near-term future are strong warming and significant increases in downward shortwave and longwave flux, with minor changes in other elements. Therefore, the HCPD-CA dataset provides extraordinary warming scenarios for assessing the impacts of future warming on many sectors (e.g., agriculture, ecological and hydrological systems) in CA. Details about changes in these meteorological elements (e.g., changes on the seasonal scale) are out of the scope of the present paper and will be presented in further studies. Systematic analyses of changes in surface air temperature, heatwaves and droughts are in Qiu et al. (2021).

# 4 Discussion

## 4.1 Uncertainties in the evaluation

To prove if considering the elevation differences between the observations and the model grids during the evaluation will give a fairer assessment of the model's skills, we take T2MEAN as an example and adjusted

the simulated T2MEAN to the elevation of the observations and then compared the adjusted T2MEAN with the observations. Here, we use the records of T2MEAN on 58 stations across CA (see the stars in Fig. 1a) as observations, which as well as the records of PREC on 52 stations (which is used in sect. 4.2, see the circles in Fig. 1a) are from Global Historical Climatology Network (GHCN) of NOAA National Climatic Data Center and have been quality controlled (Qiu et al., 2021). Note that a station is compared with the model grid on which it is located. Fig. S8 shows the SCCs and RMSEs of the simulated annual and seasonal T2MEAN over CA before and after adjusting. It is seen that the simulated T2MEAN is more consistent with the observations after vertically interpolating the model data to the elevation of the stations by the standard moist lapse rate of 6.5 ℃/km (Qiu et al., 2017). For instance, after adjusting the SCC of the simulated annual T2MEAN increases from 0.93 to 0.96 and its RMSE decreases from 2.52 to 2.25℃. This proves that the regional model's skills may be underestimated if the elevation differences between the observations and the model grids is not considered.

## 4.2 9km vs 27km

As discussed above, most of the previous RCM simulations in CA have horizontal resolutions not higher than 30km. To show the added values of using 9-km resolution in this study respect to using coarser resolutions, the evaluation metrics (SCC and RMSE) of the simulated 9-km resolution precipitation in the inner domain of the WRF model are compared with those of 27-km resolution precipitation in the outer domain (Fig. 8). As the gridded observations (CRU TS v4, and ERA5-Land) have potential limitations in depicting the climatology of precipitation in CA, the metrics are calculated based on the aforementioned 52 stations' data across CA.

Compared with the 27-km resolution data, the 9-km resolution data largely increases SCCs and reduces RMSEs, especially over the mountainous areas (see the scope of subregion "MT" in Fig. 1c). For instance, over the mountainous areas, the ensemble-mean SCC of annual precipitation increases from 0.38 to 0.58 (Fig. 8c) and the ensemble-mean RMSE of annual precipitation decreases from 1.30 to 1.14 mm/day (Fig. 8d). This highlights the necessity of improving the model resolution from ≥30km to 9km and the advantages of using the HCPD-CA dataset for researches in CA.

## 4.3 Uncertainties of the HCPD-CA dataset

With the limitation of the computational and time cost, this study used three bias-corrected GCMs from CMIP5 to do the dynamical downscaling over CA, which is an improvement relative to using a single original GCM. However, it still harbors uncertainties in the projected climate changes. As reported in the 1.5℃ special report of the Intergovernmental Panel on Climate Chane (IPCC), we are on track to exceed 1.5℃ warming

between 2030 and 2052 based on the current warming rate, and hence the near-term future projection becomes more critical to human development than that for the end of this century. Therefore, this study focuses on projected climate changes over CA in the near-term future (2031-2050). Long-term continuous (e.g., 1986-2100) regional climate projections in CA are more useful for studies in this region and will be conducted in the next stage. Land-use and land-cover (LULC) in the WRF model both in the historical and future simulations is derived from the MODIS data of 2002 (Wang et al., 2007). Dramatic changes in land-use and land-cover have happened in CA and are very likely to be ongoing in the future (Micklin, 2007;Ma et al., 2021;Chen et al., 2013;Li et al., 2019), such as the shrinking of the Aral Sea and the expansion of croplands and urbans. The land-use and land-cover changes (LULUCC) are not taken into account in our simulations, which brings uncertainties in simulating the historical climate in this area as well as projecting the climate changes in the future. A study about assessing the effects of the future LULCC on the local climate in CA is in process and the model outputs from this study will be openly published as a complement to the HCPD-CA dataset.

## 5. Data and code availability

The HCPD-CA is hosted at National Tibetan Plateau Data Center (Li et al., 2020;Pan et al., 2021) and has the DOI https://doi.org/10.11888/Meteoro.tpdc.271759 (Qiu, 2021). The files are stored in netCDF4 format and compiled using the Climate and Forecast (CF) conventions. It contains four geostatic variables and ten meteorological elements from three RCM simulations (WRF_CCSM_COR, WRF_MPI_COR, and WRF_Had_COR) for a spatial domain covering the CA region (which is consisted of Kazakhstan, Kyrgyzstan, Tajikistan, Turkmenistan, and Uzbekistan) and its surrounding areas (see the domain "D02" in Fig. 1b). The dataset covers two continuous 20-year periods, 1986-2005 and 2031-2050. Each year has 365 days (there is no leap year). We provide smaller-size (monthly and annual) files as surrogates for larger-size (daily) files. The names of the files containing the geostatic variables follow the order: [dataset name]_[variable name].nc. For example, the file name, HCPD-CA_ISLTYP.nc, represents the soil category in the HCPD-CA dataset. The names of the files containing the meteorological elements follow the order: [dataset name]_[experiment name]_[element name]_[year].[time frequency].nc. For example, the file name, HCPD-CA_WRF_CCSM_COR_T2MAX_2004.mon.nc, represents the monthly mean T2MAX of 2004 from the experiment WRF_CCSM_COR in the HCPD-CA dataset.

The WRF model is available at https://www2.mmm.ucar.edu/wrf/users/download/get_source.html. The source code to do the bias correction is available at https://rda.ucar.edu/datasets/ds316.1/#!software. The Climate Data Operators (CDO, https://code.mpimet.mpg.de/projects/cdo), Python modules (like netCDF4,

Xarray, and Numpy), and NCAR Command Languages (NCL, https://www.ncl.ucar.edu/) are recommended to do operations on the netCDF files.

# 6. Conclusions

A high-resolution (9km) projection climate dataset in CA (the HCPD-CA dataset), containing four geostatic variables and ten meteorological elements, is derived from dynamically downscaled results based on three bias-corrected GCMs (MPI-ESM-MR, CCSM4, and HadGEM2-ES) from CMIP5 for application to vulnerability, impacts, and adaption assessments in this region. The reference and future periods are 1986-2005 and 2031-2050, respectively. The carbon emission scenario is RCP4.5. The evaluation shows good quality of the data product in describing the climatology of all the meteorological elements in CA despite some systematic biases (e.g., stronger downward shortwave radiation throughout the year), which ensures the suitability of the dataset. The RCM simulations commonly suggest strong warming over CA in the near-term future, with the annual mean T2MEAN increasing by 1.62-2.02°C, and significant increase in downward shortwave and longwave flux. Changes in other elements (e. g., precipitation, relative humidity at 2m, and wind speed at 10m) are minor. The HCPD-CA dataset presented here serves as a scientific basis for assessing the impacts of climate change over CA on many sectors, especially on ecological and hydrological systems.

# Author contribution

All the authors made contributions to the conception or design of the work. YQ did the analyses and drafted the work and others revised it.

# Competing interests

The authors declare that they have no conflict of interest

# Acknowledgements

This study was supported by the Strategic Priority Research Program of Chinese Academy of Sciences (Grand No. XDA20020201) and the General Project of the National Natural Science Foundation of China (Grand No. 41875134). The work was carried out at National Supercomputer Center in Tianjin, and this research was supported by TianHe Qingsuo Project – special fund project in the field of climate, meteorology and ocean. The HCPD-CA dataset is hosted at National Tibetan Plateau Data Center (data.tpdc.ac.cn/en/).

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

# Tables

**Table 1** Geostatic variables and meteorological elements in the HCPD-CA dataset

| Name | Description | Unit |
| --- | --- | --- |
| HGT | Terrain height | m |
| LU_INDEX | Land use category | - |
| LANDMASK | Land mask (1 for land, 0 for water) | - |
| ISLTYP | Soil category | - |
| PREC | Daily precipitation | mm/day |
| T2MEAN | Daily mean temperature at 2m | K |
| T2MAX | Daily maximum temperature at 2m | K |
| T2MIN | Daily minimum temperature at 2m | K |
| RH2MEAN | Daily mean relative humidity at 2m | % |
| U10MEAN | Daily mean eastward wind at 10m | m/s |
| V10MEAN | Daily mean northward wind at 10m | m/s |
| SWD | Daily mean downwelling shortwave flux at bottom | $W/m^2$ |
| LWD | Daily mean downwelling longwave flux at bottom | $W/m^2$ |
| PSFC | Daily mean surface pressure | Pa |

**Table 2** Information about the datasets used in the study.

| Dataset | Run | Spatial Resolution | Temporal Resolution | Link |
|---|---|---|---|---|
| MPI-ESM-MR | r1i1p1 | 1.9°×1.9° | 6-hourly | https://esgf-node.llnl.gov/projects/cmip5/ |
| HadGEM2-ES | r1i1p1 | 1.3°×1.9° | 6-hourly | https://esgf-node.llnl.gov/projects/cmip5/ |
| CCSM4 | b40.[20th\RCP 4.5].track1.1de g.012.cam2.h4 | 0.9°×1.3° | 6-hourly | https://rda.ucar.edu/datasets/ds316.0/#!access |
| ERA-Interim | - | 0.75°×0.75° | Synoptic monthly means | https://apps.ecmwf.int/datasets/data/interim-full-mnth/levtype=sfc/ |
| CRU TS v4 | - | 0.5°×0.5° | monthly | https://crudata.uea.ac.uk/cru/data/hrg/cru_ts_4.00/ |
| ERA5-Land | - | 0.1°×0.1° | monthly | https://cds.climate.copernicus.eu/cdsapp#!/dataset/reanalysis-era5-land-monthly-means?tab=form |

# Figures

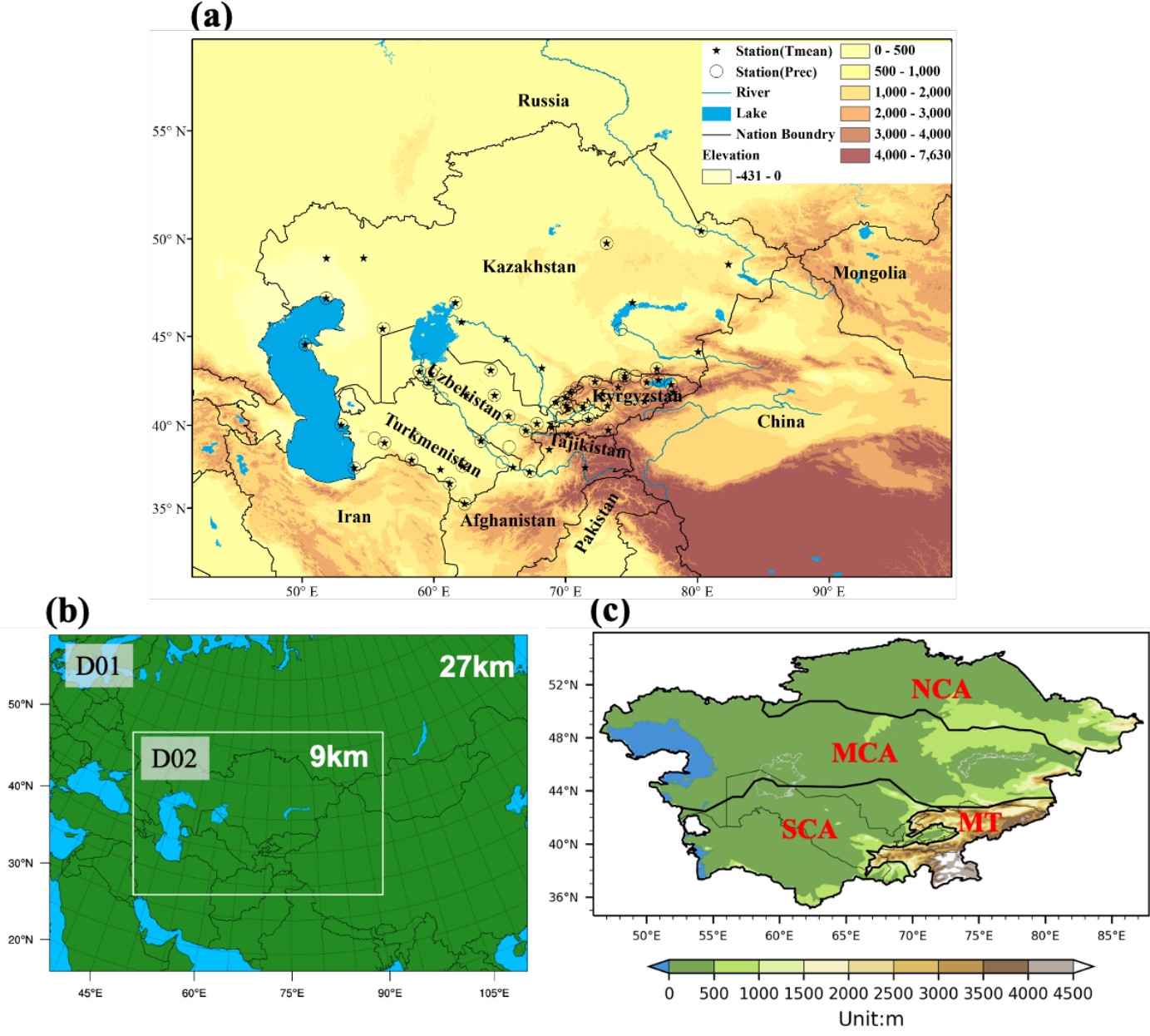

**Fig. 1** Central Asia (referred to as CA) and its surroundings (a), nested domains in the WRF model (b), and climate subregions in CA (c). In subplot a, stations with records of daily mean temperature and precipitation are marked by stars and circles, respectively. In subplot c, according to Qiu et al. (2021), the CA region is divided into four climate sub-regions: northern CA (NCA), middle CA (MCA), southern CA (SCA), and the mountainous areas (MT).

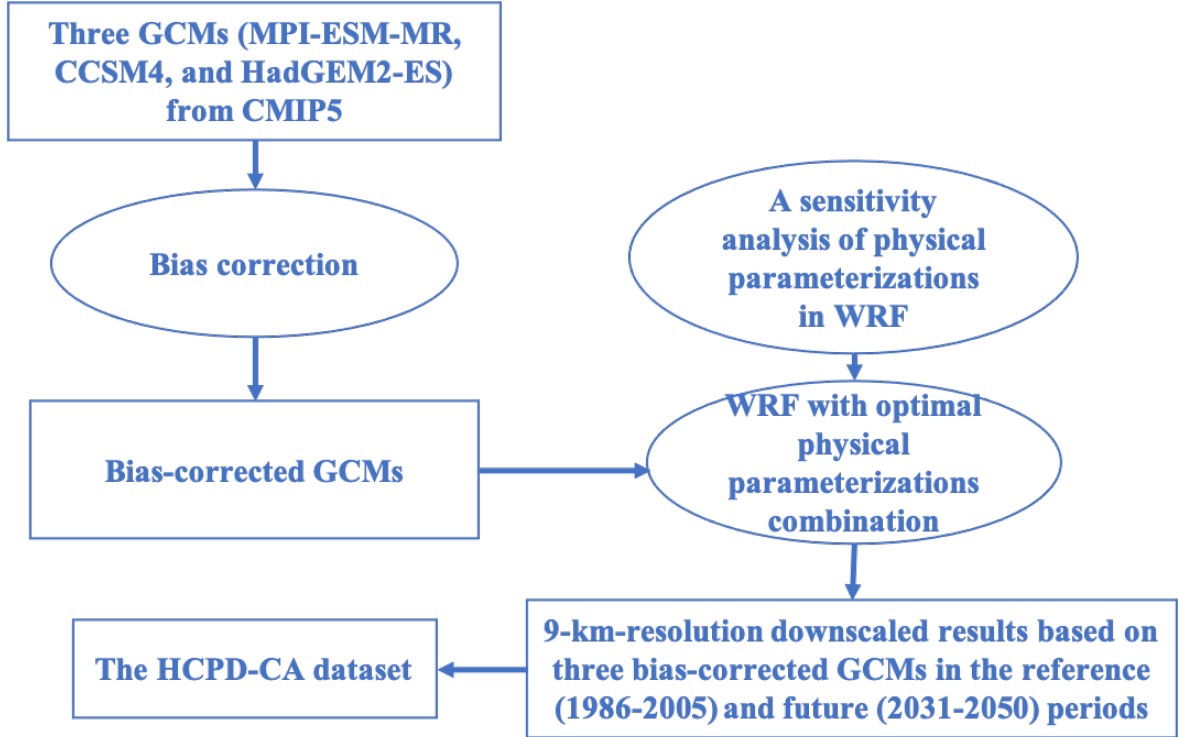

**Fig. 2** Flow chart for the HCPD-CA dataset.

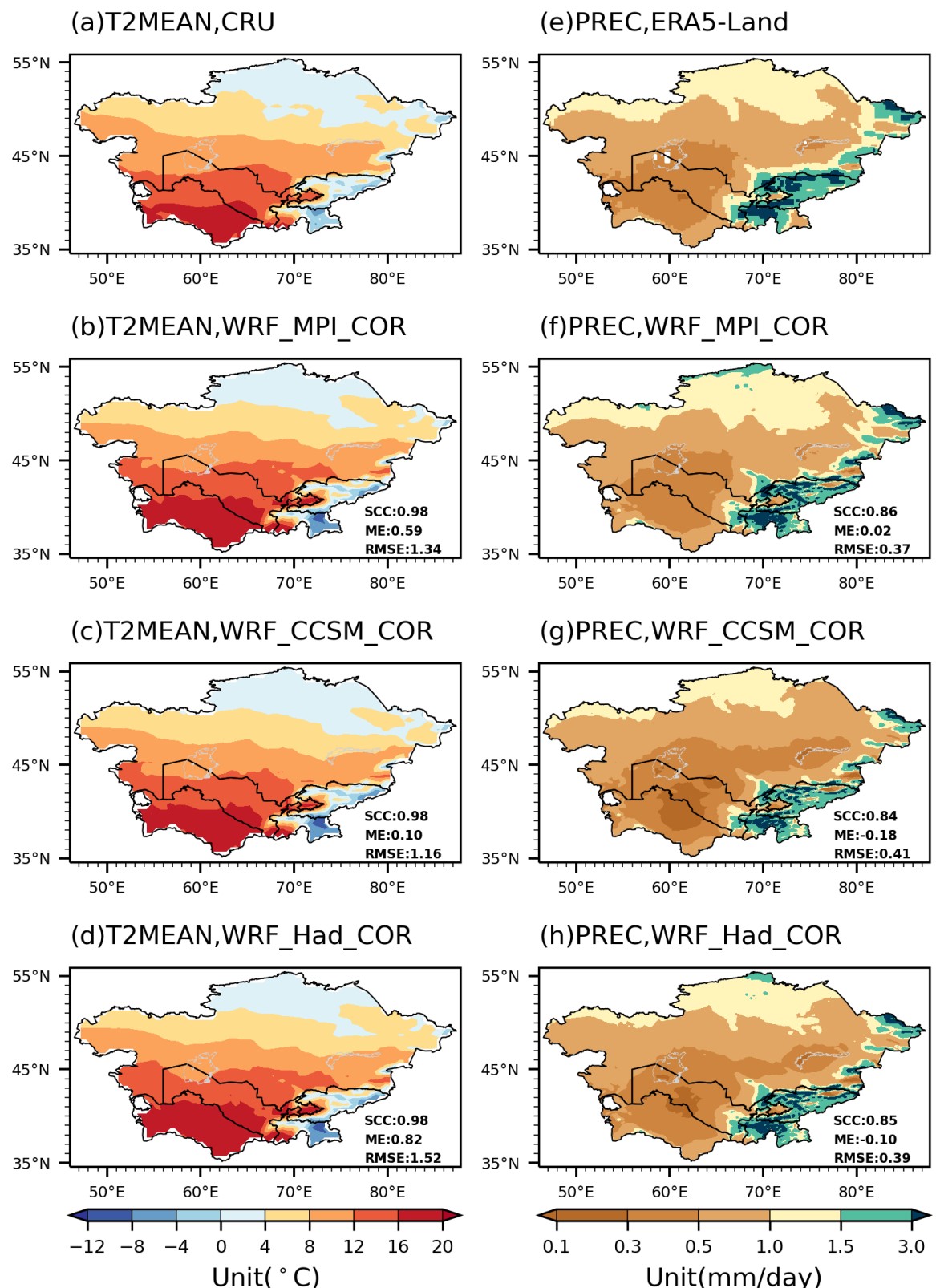

**Fig. 3** The observed and simulated annual mean T2MEAN and PREC in Central Asia during the reference period (1986-2005). The spatial correlation coefficient (SCC), mean error (ME), and root mean square error (RMSE) are listed.

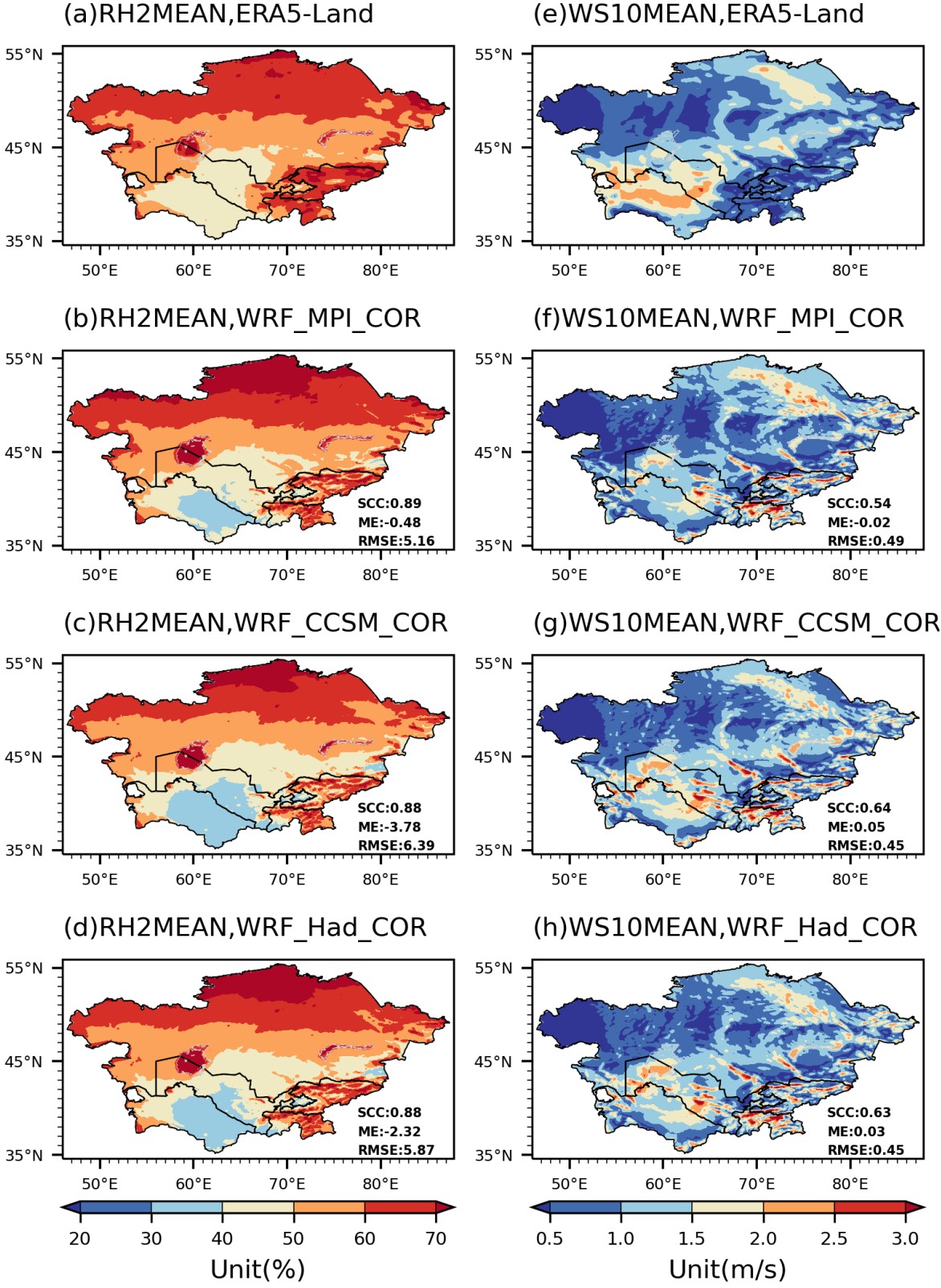

**Fig. 4** Same as **Fig. 3**, but for annual mean RH2MEAN and WS10MEAN.

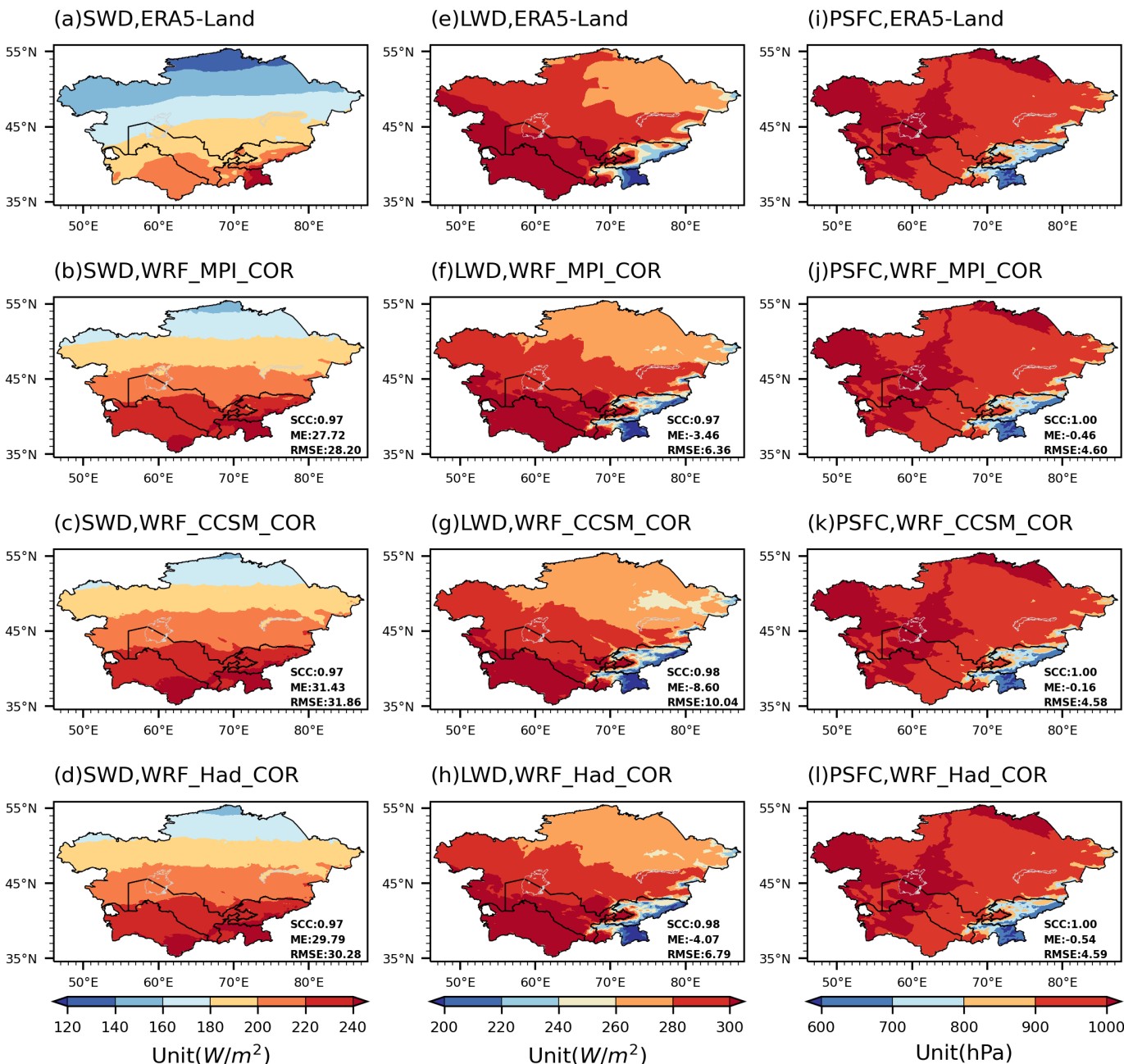

**Fig. 5** Same as **Fig. 3**, but fort annual mean SWD, LWD, and PSFC.

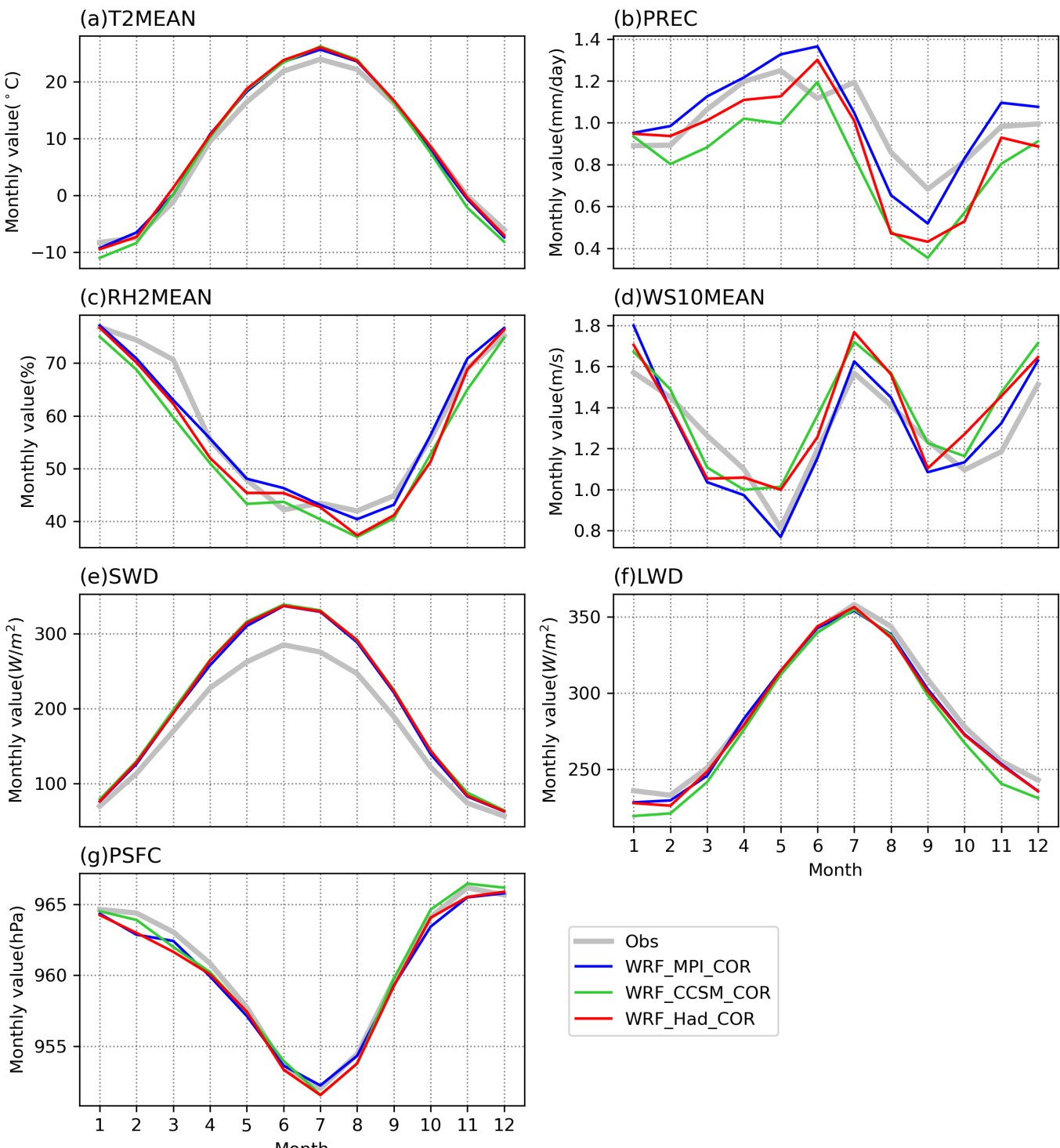

**Fig. 6** Mean annual cycle of the monthly values averaged over Central Asia in the observations and RCM simulations.

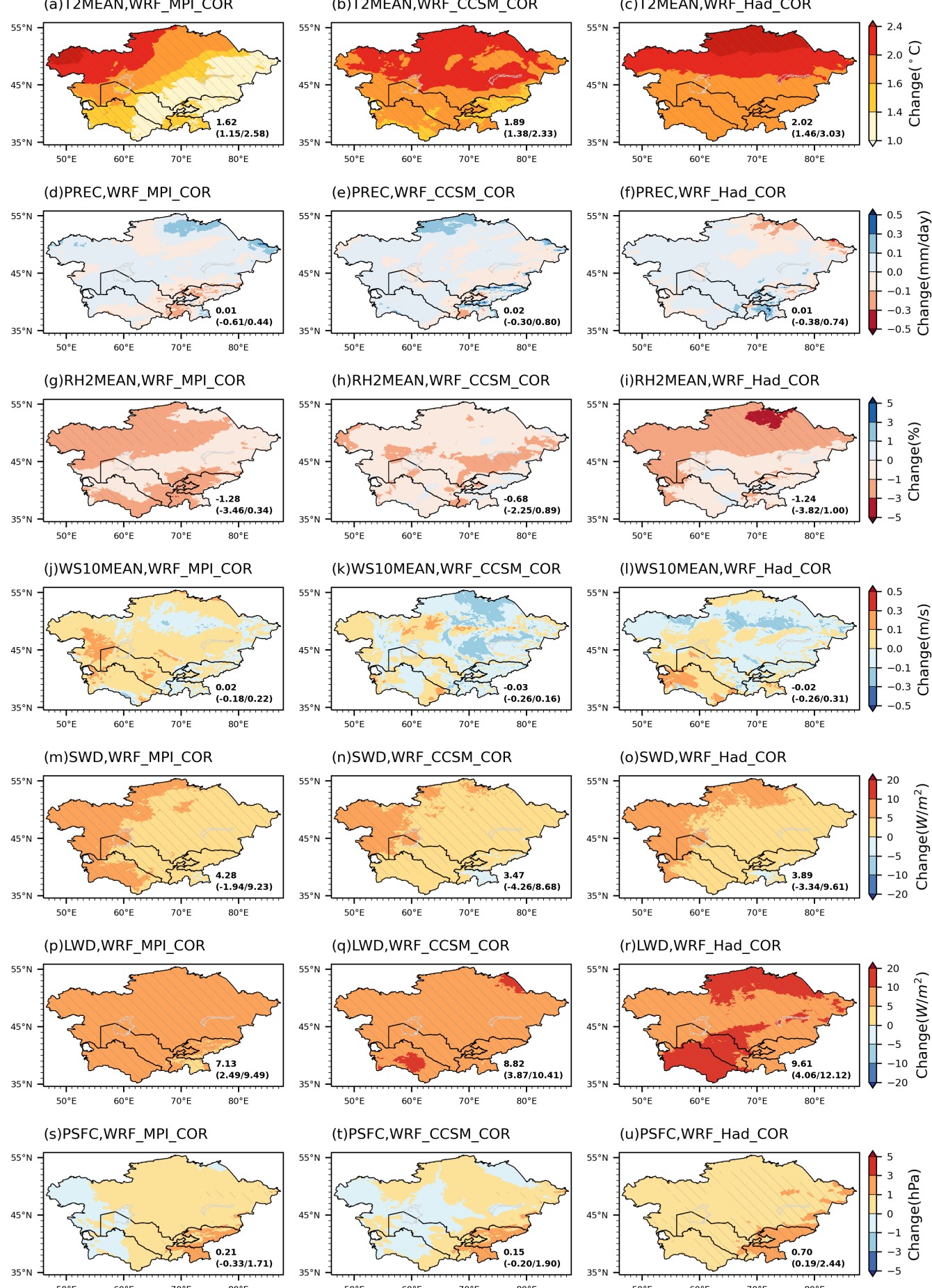

**Fig. 7** Projected changes of the annual mean values over Central Asia during 2031-2050, relative to 1986-

2005. The regional mean (upper), minimum and maximum value (in parentheses) are listed. The slashed areas indicate where the changes passed the significance test at the 95% confidence level using the two-tailed Student's t test.

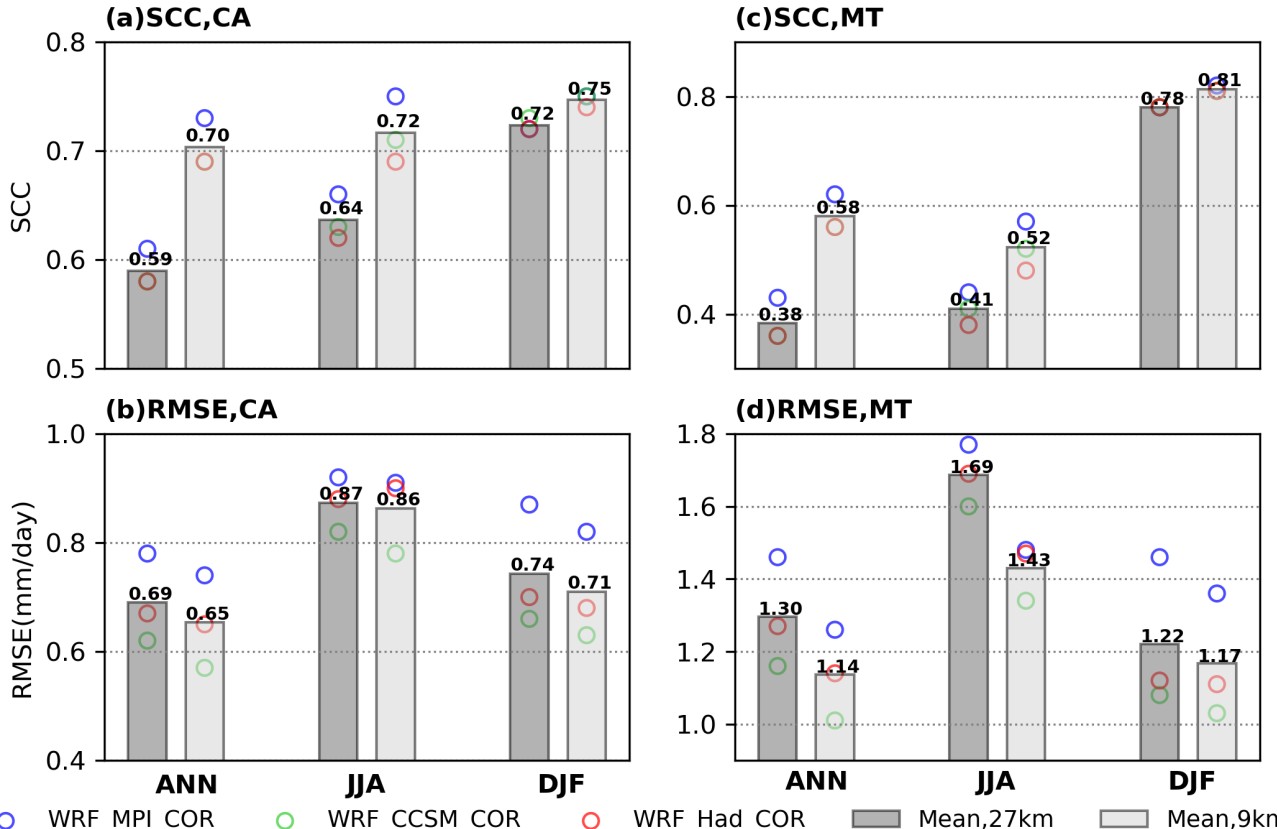

**Fig. 8** Spatial correlation coefficients (SCCs) and root mean square errors (RMSEs) of the simulated annual (ANN), summer (JJA: June-July-August), and winter (DJF: December-January-February) mean precipitation over CA and the mountainous areas (MT) in the 9-km and 27-km resolution downscaled results. The metrics are calculated based on 52 stations' data across CA.