# Peer review of "HCPD-CA High-resolution climate projection dataset in Central Asia"

_Earth System Science Data, 2021_

## Author Comment (AC4)

**RC1: 'Comment on essd-2021-361', Anonymous Referee #1, 16 Nov 2021**

Through a dynamic downscaling approach, the authors provided a high-resolution (9km) climate dataset for CA covering 10 commonly used meteorological elements. The manuscript is well organized while requiring minor revisions.

(1) Title: In fact, this study does not show ecological or hydrological applications of this high-resolution dataset. Therefore, the phrase "for ecological and hydrological applications" should be removed from the title.

Reply: We removed "for ecological and hydrological applications" from the old title. The new title is "HCPD-CA High-resolution climate projection dataset in Central Asia".

(2) Introduction: Why this method is used for downscaling of climate projections. A criticism-featured literature review on downscaling methods is needed.

Reply: We added a review of the dynamical downscaling method, to explain the necessity of using this method for projecting the local climate in Central Asia.

"Global climate models (GCMs) can describe the response of the global circulation to large-scale forcing, such as greenhouse gases and solar radiation (Giorgi, 2019). But their horizontal resolutions are too coarse to account for the effects of local-scale forcing and processes, such as complex topography, land cover distribution, and dynamical processes occurring at the mesoscale (Giorgi et al., 2016;Qiu et al., 2017;Torma et al., 2015). To obtain the accurate information on region-scale climate change, dynamical downscaling as well as statistical downscaling has been developed and widely applied in regional climate projections over main areas, like East Asia (Zou and Zhou, 2017;Bao et al., 2015;Zou and Zhou, 2016;Tang et al., 2016;Jiang et al., 2021;Guo et al., 2021;Hong et al., 2017;Ji and Kang, 2013;Jung et al., 2015), North America (Giorgi et al., 1994;Di Luca et al., 2013, 2012;Pierce et al., 2013;Racherla et al., 2012;Wang and Kotamarthi, 2015;Wang et al., 2015), and Europe (Déqué et al., 2007;Gao et al., 2006;Jacob et al., 2014;Kotlarski et al., 2014;Vautard et al., 2013;Fischer et al., 2015;Giorgi et al., 2016;Im et al., 2010;Kotlarski et al., 2015;Torma et al., 2015;Zittis et al., 2019). Some efforts have also been devoted on regional climate projection in CA with the dynamical downscaling method (Zhu et al., 2020;Ozturk et al., 2017;Mannig et al., 2013). However, their resolutions are still low (≥30km), especially for the mountainous areas in the southeast. Moreover, most of the previous RCM simulations in CA used a single GCM as the lateral boundary conditions, which harbor high uncertainties in the projected climate changes." (L31-47 in the revised MS)

(3) Model and experiments- Bias-correction technique: why did you choose these three GCMs (MPI-ESM-MR, CCSM4 and HadGEM2-ES)? The authors need to add an explanation in the manuscript.

Reply: "The reasons why we chose these three GCMs are as below: they can provide all the variables that are needed to drive the regional model; they have relatively high horizontal resolution (Table 2) among the CMIP5 models; they have fairly good performance in simulating the local temperature and precipitation in CA (see Fig. S1-4 in Qiu et al., 2021), though systematic biases exist partially due to their coarse resolution." (L100-104 in the revised MS)

*Ref: Qiu, Y., Feng, J., Yan, Z., Wang, J., and Li, Z.: High-resolution dynamical downscaling for regional climate projection in Central Asia based on bias-corrected multiple GCMs, Climate Dynamics, 10.1007/s00382-021-05934-2, 2021.*

---

## Author Comment (AC5)

**RC2: 'Comment on essd-2021-361', Anonymous Referee #2, 18 Nov 2021**

Thank the authors and editor for bringing me to this work, the bias-correction method based dynamical downscaling is a totally new technique for me and very interesting. Base on this method, the authors created high resolution historical and projected gridded climate datasets in Central Asia (CA), which are very important and useful to the target region. The manuscript is generally well written and I have only some minor points as following.

(1) I suggest to also provide the key geostatic variables from the WRF downscaling for data archive, e.g. topography, soil type, land cover type.

Reply: We have uploaded four geostatic variables from the simulation to the data repository. They are terrain height (HGT, m), land use category (LU_INDEX, 21 categories), land mask (LANDMASK, 1 for land and 0 for water), and soil category (ISLTYP, 16 categories). Accordingly, we revised the MS and updated the information about the dataset on the website of the data repository (https://doi.org/10.11888/Meteoro.tpdc.271759).

"…the HCPD-CA (High-resolution Climate Projection Dataset in CA) dataset is derived from the 9-km resolution downscaled results, which includes four geostatic (time-invariant) variables and ten meteorological elements (Table 1) that are widely used to drive ecological and hydrological models. The geostatic variables are terrain height (HGT, m), land use category (LU_INDEX, 21 categories), land mask (LANDMASK, 1 for land and 0 for water), and soil category (ISLTYP, 16 categories)." (L59-64 in the revised MS)

"The geogrid program in the WRF model is to define the simulation domains, and interpolate various terrestrial datasets to the model grids (Wang et al., 2007). First, geogrid computes the latitude, longitude, and map scale factors at every grid point. Then, it interpolates terrain height, land use category, soil category and other time-invariant data to the model grides. Global datasets of each of these fields are provided through the WRF download page (https://www2.mmm.ucar.edu/wrf/users/download/get_sources_wps_geog.html). The HCPD-CA dataset contains four of the geostatic variables. In them, the terrain height (HGT) data (Fig. S1) is from the United States Geological Survey (USGS) GTOPO30 elevation dataset, the land use category (LU_INDEX) data (Table S1 and Fig. S2) is from the Moderate Resolution Imaging Spectroradiometer (MODIS) 21 category land dataset, the soil category (ISLTYP) data (Table S2 and Fig. S3) is from the global 5-minute United Nation FAO soil category dataset, and the land mask (LANDMASK) data (Fig. S4) is calculated based on LU_INDEX with the condition that the value of a grid cell is set as 1 (0) if land (water) area at least accounts for 50%." (L86-97 in the revised MS)

"The names of the files containing the static variables follow the order: [dataset name]_[variable name].nc. For example, the file name, HCPD-CA_ISLTYP.nc, represents the soil category in the HCPD-CA dataset." (L275-277 in the revised MS)

**Table S1** Land use categories in the HCPD-CA dataset

| Land use category | Land use description |
| --- | --- |
| 1 | Evergreen needleleaf forest |
| 2 | Evergreen broadleaf forest |
| 3 | Deciduous needleleaf forest |
| 4 | Deciduous broadleaf forest |

| | |
|---|---|
| 5 | Mixed forest |
| 6 | Closed shrublands |
| 7 | Open shrublands |
| 8 | Woody savannas |
| 9 | Savannas |
| 10 | Grasslands |
| 11 | Permanent wetland |
| 12 | Croplands |
| 13 | Urban and build-up |
| 14 | Cropland/natural vegetation mosaic |
| 15 | Snow and ice |
| 16 | Barren or sparsely vegetated |
| 17 | Water |
| 18 | Wooded tundra |
| 19 | Mixed Tundra |
| 20 | Barren Tundra |
| 21 | Lakes |

**Table S2** Soil categories in the HCPD-CA dataset

| Soil category | Soil description |
|---|---|
| 1 | Sand |
| 2 | Loamy sand |
| 3 | Sandy loam |
| 4 | Silt loam |
| 5 | Silt |
| 6 | Loam |
| 7 | Sandy clay loam |
| 8 | Silty clay loam |
| 9 | Clay loam |
| 10 | Sandy clay |
| 11 | Silty clay |
| 12 | Clay |
| 13 | Organic material |
| 14 | Water |
| 15 | Bedrock |
| 16 | Other (land-ice) |

[Figure]

**Fig. S1** The terrain height in the WRF model.

[Figure]

**Fig. 2S** The main land use categories in the WRF model.

[Figure]

**Fig. 3S** The main soil categories in the WRF model.

[Figure]

**Fig. 4S** The land mask in the WRF model.

(2) Addressing the data quality is very important, especially for a data journal. As a potential user, I expect to see the spacial distribution of biases (those versus CRU, ERA5 and stations), so I can have a basic impression on the accuracy of the data when applied to any sub-region or basin for ecological and hydrological studies.

Thus, I recommend to add related figures, but you do not need to deliver in-depth scientific discussions. I also suggest to provide similar figures as figure 8 for other variables instead of only give precipitation as an example. So the readers can directly know the quality of each variable when compared with station observation.

You can add such figures as main content or even as an appendix is acceptable.

Additionally, I suggest to consider topographic elevation difference between WRF simulation and observation (CRU, ERA5, station) during comparison/evaluation, for these three variables: pressure, temperature, relative humidity. Because base on my own experience, this is very important over mountainous/complex-terrain regions.

Reply: The figures (Fig. 5S-7S) about the spatial distribution of biases have been added in the supplementary information, to help the readers have a basic impression on the accuracy of the data. To let the readers directly know the quality of each simulated variables, we made a table (Table S3) which summarizes their statistic metrics [spatial correlation coefficient (SCC), mean error (ME), and root mean square error (RMSE)] over CA and its climate subregions [northern CA (NCA), middle CA (MCA), southern CA (SCA), and the mountainous areas (MT), see their scopes in Fig. 1c]. We agree that elevation differences between the model and the observations should be taken into account when evaluating the model, to give a fairer assessment. In sect. 4.1 "Uncertainties of the evaluation", we adjusted the model simulated T2MEAN to the elevation of the station observations and then compare the adjusted model data with the observations. Results show that after adjusting the SCCs of the annual and seasonal T2MEAN over CA increases and the RMSEs decreases. This proves that the regional model's skills may be underestimated if the elevation differences between the observations and the model grids is not considered. This study is to describe the HCPD-CA dataset to the scientific community and thus we did not analyze other variables (surface pressure and relative humidity) before and after adjusting.

"The simulated annual mean T2MEAN over the very north of Kazakhstan and the Pamirs has cold bias and that over other areas generally has warm bias (Fig. 5Sa-c). However, the bias over most of CA is within -2~2°C. The annual mean RH2MEAN is generally underestimated over CA except some areas in the northern part and the Aral Sea. The RCM simulations commonly overestimate the annual mean WS10MEAN (Fig. 6Sd-f) over the mountainous areas. Stronger annual mean SWD prevails in CA in each simulation (Fig. 7Sa-c), with the mean errors (MEs) over the whole region in a range of 27.72-31.43 W/m$^2$. Meanwhile, the regional model slightly underestimates annual mean LWD (Fig. 7Sd-f). The bias in annual mean PSFC is minor over the majority of CA (Fig. 7Sg-i). Table S3 summarizes the statistic metrics [SCCs, RMSEs, and mean errors (MEs)] of all the annual mean variables over both CA and its climate subregions [northern CA (NCA), middle CA (MCA), southern CA (SCA), and the mountainous areas (MT), see their scopes in Fig. 1c], to help the readers easily check the quality of this data product in the areas they are interested." (L170-182 in the revised MS)

"To prove if considering the elevation differences between the observations and the model grids during the evaluation will give a fairer assessment of the model's skills, we take T2MEAN as an example and adjusted the simulated T2MEAN to the elevation of the observations and then compared the adjusted T2MEAN with the observations. Here, we use the records of T2MEAN on 58 stations across CA (see the stars in Fig. 1a) as observations, which as well as the records of PREC on 52 stations (which is used in sect. 4.2, see the circles in Fig. 1a) are from Global Historical Climatology Network (GHCN) of NOAA National Climatic Data Center

and have been quality controlled (Qiu et al., 2021). Note that a station is compared with the model grid on which it is located. Fig. 8S shows the SCCs and RMSEs of the simulated annual and seasonal T2MEAN over CA before and after adjusting based on the elevation differences. It is seen that the simulated T2MEAN is more consistent with the observations after vertically interpolating the model data to the elevation of the stations by the standard moist lapse rate of 6.5 °C/km (Qiu et al., 2017). For instance, after adjusting the SCC of the annual T2MEAN increases from 0.93 to 0.96 and its RMSE decreases from 2.52 to 2.25°C. This proves that the regional model's skills can be underestimated if the elevation differences between the model and the observation is not considered." (L219-231 in the revised MS)

**Table S3** Statistic metrics [spatial correlation coefficients (SCCs), mean errors (MEs), and root mean square errors (RMSEs)] of the annual mean variables in the RCM simulations over Central Asia and its climate subregions [northern CA (NCA), middle CA (MCA), southern CA (SCA), and the mountainous areas (MT), see their scopes in Fig. 1c]. The ensemble mean (first number) and the minimum and maximum member (in parentheses) are listed. The metrics are calculated based on the gridded observations (CRU TS v4 and ERA5-Land).

| Region | CA | NCA | MCA | SCA | MT |
|---|---|---|---|---|---|
| *T2MEAN (°C)* | | | | | |
| SCC | 0.98(0.98, 0.98) | 0.73(0.71,0.77) | 0.95(0.95,0.95) | 0.94(0.94,0.95) | 0.93(0.93,0.93) |
| ME | 0.50(0.10,0.82) | -0.02(-0.39,0.21) | 0.62(0.19,1.00) | 1.21(0.76,1.57) | -0.73(-0.97,-0.59) |
| RMSE | 1.34(1.16,1.52) | 1.01(0.93,1.07) | 1.06(0.81,1.29) | 1.42(1.08,1.73) | 2.64(2.55,2.69) |
| | | | | | |
| *PREC (mm/day)* | | | | | |
| SCC | 0.85(0.84,0.86) | 0.90(0.89,0.91) | 0.82(0.82,0.83) | 0.88(0.87,0.88) | 0.44(0.41,0.46) |
| ME | -0.09(-0.18,0.02) | -0.05(-0.18,0.08) | -0.09(-0.16,0.02) | -0.08(-0.15,-0.01) | -0.22(-0.39,-0.04) |
| RMSE | 0.39(0.37,0.41) | 0.22(0.20,0.25) | 0.23(0.20,0.26) | 0.25(0.22,0.28) | 1.16(1.14,1.18) |
| | | | | | |
| *RH2MEAN (%)* | | | | | |
| SCC | 0.88(0.88,0.89) | 0.87(0.87,0.88) | 0.85(0.84,0.87) | 0.71(0.70,0.73) | 0.37(0.36,0.37) |
| ME | -2.19(-3.78,-0.48) | 1.73(-0.27,3.32) | -1.38(-2.93,0.63) | -5.83(-7.21,-4.37) | -7.13(-8.31,-5.88) |
| RMSE | 5.81(5.16,6.39) | 3.92(3.13,4.47) | 3.99(3.38,4.59) | 7.57(6.30,8.78) | 10.63(9.61,11.52) |
| | | | | | |
| *WS10MEAN (m/s)* | | | | | |
| SCC | 0.60(0.54,0.64) | 0.67(0.65,0.69) | 0.80(0.74,0.83) | 0.56(0.48,0.60) | 0.03(0.03,0.03) |
| ME | 0.02(-0.02,0.05) | 0.04(-0.03,0.15) | -0.03(-0.11,0.04) | -0.14(-0.26,-0.06) | 0.77(0.76,0.78) |
| RMSE | 0.46(0.45,0.49) | 0.33(0.32,0.35) | 0.27(0.25,0.30) | 0.50(0.47,0.56) | 1.11(1.10,1.12) |
| | | | | | |
| *SWD (W/m$^2$)* | | | | | |
| SCC | 0.97(0.97,0.97) | 0.97(0.97,0.97) | 0.89(0.88,0.90) | 0.94(0.93,0.94) | 0.89(0.89,0.90) |
| ME | 29.65(27.72,31.43) | 28.32(26.52,30.21) | 29.77(27.54,31.61) | 30.09(28.28,31.93) | 31.60(30.58,32.55) |
| RMSE | 30.11(28.20,31.86) | 28.52(26.73,30.39) | 30.40(28.19,32.18) | 30.28(28.49,32.10) | 32.67(31.61,33.64) |
| | | | | | |
| *LWD (W/m$^2$)* | | | | | |
| SCC | 0.98(0.97,0.98) | 0.92(0.91,0.94) | 0.94(0.93,0.95) | 0.92(0.92,0.93) | 0.90(0.90,0.90) |
| ME | -5.38(-8.60,-3.46) | -6.87(-10.35,-4.74) | -4.62(-7.91,-2.65) | -3.42(-6.63,-1.82) | -11.54(-13.65,-9.57) |
| RMSE | 7.73(6.36,10.04) | 7.58(5.75,10.71) | 6.06(4.69,8.61) | 5.40(4.20,7.72) | 17.09(15.68,18.64) |

*PSFC (hPa)*

| | | | | | |
|---|---|---|---|---|---|
| SCC | 1.00(1.00,1.00) | 1.00(1.00,1.00) | 1.00(1.00,1.00) | 0.99(0.99,0.99) | 0.99(0.99,0.99) |
| ME | -0.39(-0.54,-0.16) | -0.33(-0.65,0.10) | -0.49(-0.64,-0.23) | -0.02(-0.28,0.14) | -1.18(-1.41,-0.98) |
| RMSE | 4.59(4.58,4.60) | 3.12(3.10,3.15) | 2.99(2.94,3.02) | 4.00(3.98,4.01) | 11.97(11.95,12.01) |

[Figure]

**Fig. 5S** The biases of the simulated annual mean T2MEAN and PREC in Central Asia during the reference period (1986-2005) relative to the observations.

[Figure]

**Fig. 6S** Same as **Fig. 5S**, but for the annual mean RH2MEAN and WS10MEAN.

[Figure]

**Fig. 7S** Same as **Fig. 5S**, but for the annual mean SWD, LWD, and PSFC.

[Figure]

**Fig. 8S** Spatial correlation coefficients (SCCs) and root mean square errors (RMSEs) of the simulated annual (ANN), summer (JJA: June-July-August), and winter (DJF: December-January-February) mean T2MEAN over CA with and without adjusting based on the elevation differences between the observations and the model grids. The metrics are calculated based on 58 stations' data across CA.

(3) Please clarify the origin of the station data during evaluation.

Reply: The stations' data is from Global Historical Climatology Network (GHCN) of NOAA National Climatic Data Center.

"Here, we use the records of T2MEAN on 58 stations across CA (see the stars in Fig. 1a) as observations, which as well as the records of PREC on 52 stations (which is used in sect. 4.2, see the circles in Fig. 1a) are from Global Historical Climatology Network (GHCN) of NOAA National Climatic Data Center and have been quality controlled (Qiu et al., 2021)." (L222-225 in the revised MS)

---

## Author Comment (AC6)

**RC3: 'Comment on essd-2021-361', Anonymous Referee #3, 07 Jan 2022**

It is my pleasure to review the paper entitled "HCPD-CA High-resolution climate projection 1 dataset in Central Asia for ecological and hydrological applications" by Qiu et al. The authors produced a high-resolution (9km) climate projection dataset over Central Asia based on the dynamically downscaled results being combined with multiple bias-corrected global climate models. This dataset is assumed to serve as a scientific basis for assessing the impacts of climate change over Central Asia on many sectors.

Given the importance of this work and its potential impact, I would like to providing the following comments for improving the manuscript.

(1) The time range in 2.2 (1981-2005/2026-2050) is different from that in 2.3 (1985-2005/2060-2050). Could you please explain or make them consistent for readable?

Reply: The bias-correction technique is developed by Bruyère et al. (2014). They produced the bias-corrected CCSM4 outputs (DOI: https://doi.org/10.5065/D6DJ5CN4) with a 25-year base period (1981-2005) during the bias correction. In this study, we produced the bias-corrected MPI-ESM-MR and HadGEM2-ES outputs with the same base period as them. The base period used during the bias correction is not necessary to be consistent with the reference period (1986-2005) of the RCM simulations.

"The bias-corrected CCSM4 outputs (DOI: https://doi.org/10.5065/D6DJ5CN4) is produced by Bruyère et al. (2014) with a 25-year base period (1981-2005) during the bias correction. In this study, we produced the bias-corrected MPI-ESM-MR and HadGEM2-ES outputs with the same base period as them. Note that the base period used during the bias correction is not necessary to be consistent with the reference period (1986-2005) of the RCM simulations." (L127-130 in the revised MS)

*Ref: Bruyère, C. L., Done, J. M., Holland, G. J., and Fredrick, S.: Bias corrections of global models for regional climate simulations of high-impact weather, Climate Dynamics, 43, 1847-1856, 10.1007/s00382-013-2011-6, 2014.*

(2) Fig. 2 should be explained in detail, especially for the right side of the figure. There are some sub-questions are occurred from this figure:

- How to achieve the action named "WRF with the optional combination of physical schemes" with the input from Bias-corrected GCMs and Observation data?
- The term "WRF with the optional combination of physical schemes" seems to run WRF model again, however, WRF model has been executed in previous step. This term can be renamed properly.
- Is the spatial resolution of Bias-corrected GCMs same to the one of the WRF model output? If not, how to match them?

Reply: I explained Fig. 2 in the revised MS.

"Fig. 2 shows the flow chart to produce the HCPD-CA dataset. The procedure can be divided into four steps. First, multiple-source observational data is used to evaluate the WRF model with different combinations of physical schemes and then we found the optimal combination of physical schemes for the WRF model. Second,

the original GCMs are bias corrected and the bias-corrected GCMs are used to drive the WRF model with the optimal combination of physical schemes. Third, we conducted the dynamical downscaling and produced 9-km resolution downscaled results. At last, the HCPD-CA dataset with certain variables and standard file formats is derived from the downscaled results." (L141-147 in the revised MS) As the driving data, the bias-corrected GCMs have much coarser spatial resolution than the WRF model outputs.

(6) In Fig.3, the observational data for temperature is CRU data, the ones for precipitation and other variables are ERA data. Could you please give more reasonable explanation?

Reply: We found the CRU TS v4 dataset generally has good performance to describe the climatology of surface air temperature over CA (Qiu et al., 2021). Thus, we used it to evaluate the simulated T2MEAN/T2MAX/T2MIN. Because the rain-gauge-observation merged in the CRU TS v4 dataset is sparse and unevenly distributed over CA, the precipitation data in it has limitations in depicting the climatology of precipitation in CA, especially over the mountainous areas. In addition, the CRU TS v4 dataset does not have other variables (e.g., relative humidity, wind, and shortwave and longwave radiation). As a results, we used the ERA5-Land dataset to evaluate precipitation and other variables.

*Ref: Qiu Y., Feng, J., Yan, Z., Wang, J., and Li, Z.: High-resolution dynamical downscaling for regional climate projection in Central Asia based on bias-corrected multiple GCMs, Climate Dynamics, 10.1007/s00382-021-05934-2, 2021.*

(7) It is recommended to do more comparisons between the simulation from control experiment and the one being combing Bias-correction GCMs, and to show the improvement by introducing the Bias-correction GCMs.

Reply: "In a recent study (Qiu et al., 2021), we conducted the sensitivity experiments of using the bias-correction technique, to quantify its contribution to improving the RCM simulation. The results show that using the bias-correction technique largely reduced the biases in the simulated annual and seasonal precipitation over CA respect to not using it and slightly improved the model's skill in simulating the spatial pattern of precipitation (see Fig. 4 in Qiu et al., 2021)." (L123-127 in the revised MS)

*Ref: Qiu Y., Feng, J., Yan, Z., Wang, J., and Li, Z.: High-resolution dynamical downscaling for regional climate projection in Central Asia based on bias-corrected multiple GCMs, Climate Dynamics, 10.1007/s00382-021-05934-2, 2021.*

(8) Please use the data doi linkage instead of the uuid linkage for accessing the dataset in Lines 22-23 and Lines 187-188.

Reply: Revised.

"It has the DOI https://doi.org/10.11888/Meteoro.tpdc.271759 (Qiu, 2021)." (L23-24 in the revised MS)

"The HCPD-CA has the DOI https://doi.org/10.11888/Meteoro.tpdc.271759 (Qiu, 2021)." (L267 in the revised MS)

---

## Author Comment (AC7)

**RC4: 'Comment on essd-2021-361', Anonymous Referee #4, 10 Jan 2022**

This manuscript describes a high-resolution climate projection dataset in central Asia, which could be potentially useful for hydrological or ecological applications. More proofs of its reliability and confidence levels are necessary to present before it could be used in the consequent applications.

(1) The horizontal resolution of 9-km was used in this study. My question is why 9-km? 9-km is an awaked resolution not large as precedent quarter degree, not small as convection-permitting modeling. At this resolution, whether the cumulus parametrization is used or not is still an open question.

Reply: Here, we carried out a study that involves the dynamical downscaling of multiple bias-corrected GCMs for the CA region with an unprecedented horizontal resolution of 9km. The 9-km resolution is much higher than those ($\geq$30km) of the previous regional climate projections in CA. With the limitation of the computational and time cost, we did not use a higher resolution (e.g., 3-5km) that allows convection-permitting modeling.

(2) The bias-corrected GCMs were used as forcing in this study. What does the simulation performance look like if the bias correction was not used? The previous study also claimed that similar performances were obtained in the Tibetan Plateau using either reanalysis or GCMs as forcing in the historical period. Major differences only occur in the temporal changes or linear trends. Almost the same historical simulations were also presented in figures 3-6 using three different forcings GCMs in this study. Therefore, it would be good to present the differences between using bias-corrected GCM or not.

Reply: "In a recent study (Qiu et al., 2021), we conducted the sensitivity experiments of using the bias-correction technique, to quantify its contribution to improving the RCM simulation. The results show that using the bias-correction technique largely reduced the biases in the simulated annual and seasonal precipitation over CA respect to not using it and slightly improved the model's skill in simulating the spatial pattern of precipitation (see Fig. 4 in Qiu et al., 2021)." (L123-127 in the revised MS)

Ref: Qiu Y., Feng, J., Yan, Z., Wang, J., and Li, Z.: High-resolution dynamical downscaling for regional climate projection in Central Asia based on bias-corrected multiple GCMs, Climate Dynamics, 10.1007/s00382-021-05934-2, 2021.

(3) The future changes in 2031-2050 compared to 1986-2005 are studied. It is neither the end of this century 2100 nor the target year of zero carbon emission 2060. It would be better to describe the importance of this period in the future in central Asia. In addition, all the future changes are based on the model. Suggest adding historical changes or linear trends to solid the credibility or reliability of future changes.

Reply: "As reported in the 1.5°C special report of the Intergovernmental Panel on Climate Chane (IPCC), we are on track to exceed 1.5°C warming between 2030 and 2052 based on the current warming rate, and hence the near-term future projection becomes more critical to human development than that for the end of this century. Therefore, this study focuses on projected climate changes over CA in the near-term future (2031-2050). Long-term continuous (e.g., 1986-2100) regional climate projections in CA are more useful for studies in this region and will be conducted in the next stage." (L248-256 in the revised MS) We discussed our findings based on the RCM simulations with the previous studies in a recent study (Qiu et al., 2021). For instance, enhanced warming projected in many mountains in the world is not found in CA, which is consistent with the study based on the reanalysis datasets during the past (Hu et al, 2014). Stronger warming is detected

in the north part of Central Asia, consistent with the previous regional climate projections (Mannig et al., 2013;Ozturk et al., 2017;Peng et al., 2019). A comprehensive comparison between the future and historical changes based on multiple methods and multi-source data by our group is in process.

Ref:

Hu, Z., Zhang, C., Hu, Q., and Tian, H.: Temperature Changes in Central Asia from 1979 to 2011 Based on Multiple Datasets, 27, 1143-1167, 10.1175/jcli-d-13-00064.1, 2014.

Mannig, B., Müller, M., Starke, E., Merkenschlager, C., Mao, W., Zhi, X., Podzun, R., Jacob, D., and Paeth, H.: Dynamical downscaling of climate change in Central Asia, Global and Planetary Change, 110, 26-39, https://doi.org/10.1016/j.gloplacha.2013.05.008, 2013.

Ozturk, T., Turp, M. T., Türkeş, M., and Kurnaz, M. L.: Projected changes in temperature and precipitation climatology of Central Asia CORDEX Region 8 by using RegCM4.3.5, Atmospheric Research, 183, 296-307, https://doi.org/10.1016/j.atmosres.2016.09.008, 2017.

Peng, D., Zhou, T., Zhang, L., and Zou, L. J. C. D.: Detecting human influence on the temperature changes in Central Asia, 53, 4553-4568, 2019.

Qiu, Y., Feng, J., Yan, Z., Wang, J., and Li, Z.: High-resolution dynamical downscaling for regional climate projection in Central Asia based on bias-corrected multiple GCMs, Climate Dynamics, 10.1007/s00382-021-05934-2, 2021.

---

## Referee Report (RR1)

The authors have done an excellent job in revising the manuscript. I have no further questions and recommend acceptance of the manuscript.

---

## Author Response (AR3)

We appreciate the Topical Editor and the anonymous referees for your constructive comments on our manuscript. According to the comments, we have revised the manuscript. Below are our replies with a clear and easy-to-follow sequence: (1) comments from referees, (2) authors' reply (colored by blue), and authors' changes in manuscript (colored by blue, in parentheses, and with line numbers of the revised MS using track changes).

**Topical editor decision: Publish subject to minor revisions (review by editor)**

This paper produced a high resolution (9 km) climate projection dataset over the Central Asia, which will serve as a scientific basis for assessing the impacts of climate changes for this region. I think this paper can be acceptable for publication after the following minor issues can be addressed:

(1) In Fig. 2, the term "WRF sensitive analysis" is easier for readers to understand than those "Multiple combinations of physical schemes" + "WRF" + "Observational data"; and the term "WRF with the optional combination of physical schemes" seems not right, maybe you can describe the term as "WRF with the combinational optimization of physical schemes".

Reply: Fig. 2 is revised, as well as the relevant sentences in the MS.

"Its physical schemes are set based on our previous work about the sensitivity analysis of physical parameterizations in the WRF model for local climate simulations in CA (Wang et al., 2020). Details about the optimal physical schemes are in Qiu et al. (2021)." (L74-77)

"First, a sensitivity analysis of physical parameterizations in the WRF model was done and then we identified the optimal physical parameterizations combination for WRF for regional climate studies over CA. Second, the original GCMs are bias corrected and the bias-corrected GCMs are used to drive the WRF model with the optimal physical schemes." (L136-141)

[Figure]

Fig. 2 Flow chart for the HCPD-CA dataset.

(2) Since you have put the data in National Tibetan Plateau/Third Pole Environment Data Center, you are welcome to cite the relevant introduction papers into the articles: https://doi.org/10.1175/BAMS-D-21-0004.1 and https://doi.org/10.1175/BAMS-D-19-0280.1

Reply: We added the relevant references.

"The HCPD-CA is hosted at National Tibetan Plateau Data Center (Li et al., 2020;Pan et al., 2021)…" (L257)

**RC1: 'Comment on essd-2021-361', Anonymous Referee #1, 16 Nov 2021**

Through a dynamic downscaling approach, the authors provided a high-resolution (9km) climate dataset for CA covering 10 commonly used meteorological elements. The manuscript is well organized while requiring minor revisions.

(1) Title: In fact, this study does not show ecological or hydrological applications of this high-resolution dataset. Therefore, the phrase "for ecological and hydrological applications" should be removed from the title.

Reply: We removed "for ecological and hydrological applications" from the old title. The new title is "HCPD-CA High-resolution climate projection dataset in Central Asia" (L1).

(2) Introduction: Why this method is used for downscaling of climate projections. A criticism-featured literature review on downscaling methods is needed.

Reply: We added a review of the dynamical downscaling method, to explain the necessity of using this method for projecting the local climate in Central Asia.

"Global climate models (GCMs) can describe the response of the global circulation to large-scale forcing, such as greenhouse gases and solar radiation (Giorgi, 2019). But their horizontal resolutions are too coarse to account for the effects of local-scale forcing and processes, such as complex topography, land cover distribution, and dynamical processes occurring at the mesoscale (Giorgi et al., 2016;Qiu et al., 2017;Torma et al., 2015). To obtain the accurate information on region-scale climate change, dynamical downscaling has been developed and widely applied in regional climate projections over many areas, like East Asia (Zou and Zhou, 2017;Bao et al., 2015;Zou and Zhou, 2016;Tang et al., 2016;Jiang et al., 2021;Guo et al., 2021;Hong et al., 2017;Ji and Kang, 2013;Jung et al., 2015), North America (Giorgi et al., 1994;Di Luca et al., 2013, 2012;Pierce et al., 2013;Racherla et al., 2012;Wang and Kotamarthi, 2015;Wang et al., 2015), and Europe (Déqué et al., 2007;Gao et al., 2006;Jacob et al., 2014;Kotlarski et al., 2014;Vautard et al., 2013;Fischer et al., 2015;Giorgi et al., 2016;Im et al., 2010;Kotlarski et al., 2015;Torma et al., 2015;Zittis et al., 2019). Some efforts have also been devoted on regional climate projection in CA with the dynamical downscaling method (Zhu et al., 2020;Ozturk et al., 2017;Mannig et al., 2013). However, their resolutions are still low (≥30km), especially for the mountainous areas in the southeast. Moreover, most of the previous RCM simulations in CA used a single GCM as the lateral boundary conditions, which harbor high uncertainties in the projected climate changes." (L36-55)

(3) Model and experiments- Bias-correction technique: why did you choose these three GCMs (MPI-ESM-MR, CCSM4 and HadGEM2-ES)? The authors need to add an explanation in the manuscript.

Reply: "The reasons why we chose these three GCMs are as below: they can provide all the variables that are needed to drive the regional model; they have relatively high horizontal resolution (Table 2) among the CMIP5 models; they have fairly good performance in simulating the local temperature and precipitation in CA (see Fig. S1 and S3 in Qiu et al., 2021), though systematic biases exist partially due to their coarse resolutions." (L117-122)

*Ref: Qiu, Y., Feng, J., Yan, Z., Wang, J., and Li, Z.: High-resolution dynamical downscaling for regional climate projection in Central Asia based on bias-corrected multiple GCMs, Climate Dynamics, 10.1007/s00382-021-05934-2, 2021.*

**RC2: 'Comment on essd-2021-361', Anonymous Referee #2, 18 Nov 2021**

Thank the authors and editor for bringing me to this work, the bias-correction method based dynamical downscaling is a totally new technique for me and very interesting. Base on this method, the authors created high resolution historical and projected gridded climate datasets in Central Asia (CA), which are very

important and useful to the target region. The manuscript is generally well written and I have only some minor points as following.

(1) I suggest to also provide the key geostatic variables from the WRF downscaling for data archive, e.g. topography, soil type, land cover type.

Reply: We have uploaded four geostatic variables from the simulation to the data repository. They are terrain height (HGT, m), land use category (LU_INDEX, 21 categories), land mask (LANDMASK, 1 for land and 0 for water), and soil category (ISLTYP, 16 categories). Accordingly, we revised the manuscript and updated the information about the dataset on the website of the data repository (https://doi.org/10.11888/Meteoro.tpdc.271759).

"…the HCPD-CA (High-resolution Climate Projection Dataset in CA) dataset is derived from the 9-km resolution downscaled results, which includes four geostatic (time-invariant) variables and ten meteorological elements (Table 1) that are widely used to drive ecological and hydrological models. The geostatic variables are terrain height (HGT, m), land use category (LU_INDEX, 21 categories), land mask (LANDMASK, 1 for land and 0 for water), and soil category (ISLTYP, 16 categories)." (L67-73)

"The geogrid program in the WRF model is to define the simulation domains, and interpolate various terrestrial datasets to the model grids (Wang et al., 2007). First, geogrid computes the latitude, longitude, and map scale factors at every grid point. Then, it interpolates terrain height, land use category, soil category and other time-invariant data to the model grides. Global datasets of each of these fields are provided through the WRF download page (https://www2.mmm.ucar.edu/wrf/users/download/get_sources_wps_geog.html). The HCPD-CA dataset contains four of the geostatic variables. In them, the terrain height (HGT) data (Fig. S1) is from the United States Geological Survey (USGS) GTOPO30 elevation dataset, the land use category (LU_INDEX) data (Table S1 and Fig. S2) is from the Moderate Resolution Imaging Spectroradiometer (MODIS) 21 category land dataset, the soil category (ISLTYP) data (Table S2 and Fig. S3) is from the global 5-minute United Nation FAO soil category dataset, and the land mask (LANDMASK) data (Fig. S4) is calculated based on LU_INDEX with the condition that the value of a grid cell is set as 1 (0) if land (water) area at least accounts for 50%." (L99-113)

"The names of the files containing the geostatic variables follow the order: [dataset name]_[variable name].nc. For example, the file name, HCPD-CA_ISLTYP.nc, represents the soil category in the HCPD-CA dataset." (L318-321)

**Table S1** Land use categories in the HCPD-CA dataset

| Land use category | Land use description |
| --- | --- |
| 1 | Evergreen needleleaf forest |
| 2 | Evergreen broadleaf forest |
| 3 | Deciduous needleleaf forest |
| 4 | Deciduous broadleaf forest |
| 5 | Mixed forest |
| 6 | Closed shrublands |
| 7 | Open shrublands |
| 8 | Woody savannas |

| | |
|---|---|
| 9 | Savannas |
| 10 | Grasslands |
| 11 | Permanent wetland |
| 12 | Croplands |
| 13 | Urban and build-up |
| 14 | Cropland/natural vegetation mosaic |
| 15 | Snow and ice |
| 16 | Barren or sparsely vegetated |
| 17 | Water |
| 18 | Wooded tundra |
| 19 | Mixed Tundra |
| 20 | Barren Tundra |
| 21 | Lakes |

**Table S2** Soil categories in the HCPD-CA dataset

| Soil category | Soil description |
|---|---|
| 1 | Sand |
| 2 | Loamy sand |
| 3 | Sandy loam |
| 4 | Silt loam |
| 5 | Silt |
| 6 | Loam |
| 7 | Sandy clay loam |
| 8 | Silty clay loam |
| 9 | Clay loam |
| 10 | Sandy clay |
| 11 | Silty clay |
| 12 | Clay |
| 13 | Organic material |
| 14 | Water |
| 15 | Bedrock |
| 16 | Other (land-ice) |

[Figure]

**Fig. S1** The terrain height in the WRF model.

[Figure]

**Fig. S2** The main land use categories in the WRF model.

[Figure]

**Fig. S3** The main soil categories in the WRF model.

[Figure]

**Fig. S4** The land mask in the WRF model.

(2) Addressing the data quality is very important, especially for a data journal. As a potential user, I expect to see the spatial distribution of biases (those versus CRU, ERA5 and stations), so I can have a basic impression on the accuracy of the data when applied to any sub-region or basin for ecological and hydrological studies.

Thus, I recommend to add related figures, but you do not need to deliver in-depth scientific discussions. I also suggest to provide similar figures as figure 8 for other variables instead of only give precipitation as an example. So the readers can directly know the quality of each variable when compared with station observation. You can add such figures as main content or even as an appendix is acceptable. Additionally, I suggest to consider topographic elevation difference between WRF simulation and observation (CRU, ERA5, station) during comparison/evaluation, for these three variables: pressure, temperature, relative humidity. Because base on my own experience, this is very important over mountainous/complex-terrain regions.

Reply: The figures (Fig. 5S, 6S and 7S) about the spatial distribution of biases have been added in the supplementary information, to help the readers have a basic impression on the accuracy of the data. To let the readers directly know the quality of each simulated variables, we made a table (Table S3) which summarizes their statistic metrics [spatial correlation coefficient (SCC), mean error (ME), and root mean square error (RMSE)] over CA and its climate subregions [northern CA (NCA), middle CA (MCA), southern CA (SCA), and the mountainous areas (MT), see their scopes in Fig. 1c]. We agree that elevation differences between the model and the observations should be taken into account when evaluating the model, to give a fairer assessment. In sect. 4.1 "Uncertainties in the evaluation", we adjusted the model simulated T2MEAN to the elevation of the station observations and then compare the adjusted model data with the observations. Results show that after adjusting the SCCs of the annual and seasonal T2MEAN over CA increases and the RMSEs decreases. This proves that the regional model's skills may be underestimated if the elevation differences between the model grids and the observations is not considered. This study is to describe the HCPD-CA dataset to the scientific community and thus we did not analyze other variables (surface pressure and relative humidity) before and after adjusting.

"The simulated annual mean T2MEAN over the very north of Kazakhstan and the Pamirs has cold bias and that over other areas generally has warm bias (Fig. 5Sa-c). However, the bias over most of CA is within -2~2°C. The annual mean RH2MEAN is generally underestimated over CA except some areas in the northern part and the Aral Sea. The RCM simulations commonly overestimate the annual mean WS10MEAN (Fig. 6Sd-f) over the mountainous areas. Stronger annual mean SWD prevails in CA in each simulation (Fig. 7Sa-c), with the mean errors (MEs) over the whole region in a range of 27.72-31.43 W/m². Meanwhile, the regional model slightly underestimates annual mean LWD (Fig. 7Sd-f). The bias in annual mean PSFC is minor over the majority of CA (Fig. 7Sg-i). Table S3 summarizes the statistic metrics [SCCs, RMSEs, and mean errors (MEs)] of all the annual mean variables over both CA and its climate subregions [northern CA (NCA), middle CA (MCA), southern CA (SCA), and the mountainous areas (MT), see their scopes in Fig. 1c], to help the readers easily check the quality of this data product in the areas they are interested." (L197-211)

"To prove if considering the elevation differences between the observations and the model grids during the evaluation will give a fairer assessment of the model's skills, we take T2MEAN as an example and adjusted the simulated T2MEAN to the elevation of the observations and then compared the adjusted T2MEAN with the observations. Here, we use the records of T2MEAN on 58 stations across CA (see the stars in Fig. 1a) as observations, which as well as the records of PREC on 52 stations (which is used in sect. 4.2, see the circles in Fig. 1a) are from Global Historical Climatology Network (GHCN) of NOAA National Climatic Data Center and have been quality controlled (Qiu et al., 2021). Note that a station is compared with the model grid on which it is located. Fig. 8S shows the SCCs and RMSEs of the simulated annual and seasonal T2MEAN over CA before and after adjusting based on the elevation differences. It is seen that the simulated T2MEAN is

more consistent with the observations after vertically interpolating the model data to the elevation of the stations by the standard moist lapse rate of 6.5 °C/km (Qiu et al., 2017). For instance, after adjusting the SCC of the annual T2MEAN increases from 0.93 to 0.96 and its RMSE decreases from 2.52 to 2.25°C. This proves that the regional model's skills can be underestimated if the elevation differences between the model and the observation is not considered." (L252-267)

**Table S3** Statistic metrics [spatial correlation coefficients (SCCs), mean errors (MEs), and root mean square errors (RMSEs)] of the annual mean variables in the RCM simulations over Central Asia and its climate subregions [northern CA (NCA), middle CA (MCA), southern CA (SCA), and the mountainous areas (MT), see their scopes in Fig. 1c]. The ensemble mean (first number) and the minimum and maximum member (in parentheses) are listed. The metrics are calculated based on the gridded observations (CRU TS v4 and ERA5-Land).

| Region | CA | NCA | MCA | SCA | MT |
|---|---|---|---|---|---|
| *T2MEAN (°C)* | | | | | |
| SCC | 0.98(0.98, 0.98) | 0.73(0.71,0.77) | 0.95(0.95,0.95) | 0.94(0.94,0.95) | 0.93(0.93,0.93) |
| ME | 0.50(0.10,0.82) | -0.02(-0.39,0.21) | 0.62(0.19,1.00) | 1.21(0.76,1.57) | -0.73(-0.97,-0.59) |
| RMSE | 1.34(1.16,1.52) | 1.01(0.93,1.07) | 1.06(0.81,1.29) | 1.42(1.08,1.73) | 2.64(2.55,2.69) |
| *PREC (mm/day)* | | | | | |
| SCC | 0.85(0.84,0.86) | 0.90(0.89,0.91) | 0.82(0.82,0.83) | 0.88(0.87,0.88) | 0.44(0.41,0.46) |
| ME | -0.09(-0.18,0.02) | -0.05(-0.18,0.08) | -0.09(-0.16,0.02) | -0.08(-0.15,-0.01) | -0.22(-0.39,-0.04) |
| RMSE | 0.39(0.37,0.41) | 0.22(0.20,0.25) | 0.23(0.20,0.26) | 0.25(0.22,0.28) | 1.16(1.14,1.18) |
| *RH2MEAN (%)* | | | | | |
| SCC | 0.88(0.88,0.89) | 0.87(0.87,0.88) | 0.85(0.84,0.87) | 0.71(0.70,0.73) | 0.37(0.36,0.37) |
| ME | -2.19(-3.78,-0.48) | 1.73(-0.27,3.32) | -1.38(-2.93,0.63) | -5.83(-7.21,-4.37) | -7.13(-8.31,-5.88) |
| RMSE | 5.81(5.16,6.39) | 3.92(3.13,4.47) | 3.99(3.38,4.59) | 7.57(6.30,8.78) | 10.63(9.61,11.52) |
| *WS10MEAN (m/s)* | | | | | |
| SCC | 0.60(0.54,0.64) | 0.67(0.65,0.69) | 0.80(0.74,0.83) | 0.56(0.48,0.60) | 0.03(0.03,0.03) |
| ME | 0.02(-0.02,0.05) | 0.04(-0.03,0.15) | -0.03(-0.11,0.04) | -0.14(-0.26,-0.06) | 0.77(0.76,0.78) |
| RMSE | 0.46(0.45,0.49) | 0.33(0.32,0.35) | 0.27(0.25,0.30) | 0.50(0.47,0.56) | 1.11(1.10,1.12) |
| *SWD (W/m$^2$)* | | | | | |
| SCC | 0.97(0.97,0.97) | 0.97(0.97,0.97) | 0.89(0.88,0.90) | 0.94(0.93,0.94) | 0.89(0.89,0.90) |
| ME | 29.65(27.72,31.43) | 28.32(26.52,30.21) | 29.77(27.54,31.61) | 30.09(28.28,31.93) | 31.60(30.58,32.55) |
| RMSE | 30.11(28.20,31.86) | 28.52(26.73,30.39) | 30.40(28.19,32.18) | 30.28(28.49,32.10) | 32.67(31.61,33.64) |
| *LWD (W/m$^2$)* | | | | | |
| SCC | 0.98(0.97,0.98) | 0.92(0.91,0.94) | 0.94(0.93,0.95) | 0.92(0.92,0.93) | 0.90(0.90,0.90) |
| ME | -5.38(-8.60,-3.46) | -6.87(-10.35,-4.74) | -4.62(-7.91,-2.65) | -3.42(-6.63,-1.82) | -11.54(-13.65,-9.57) |
| RMSE | 7.73(6.36,10.04) | 7.58(5.75,10.71) | 6.06(4.69,8.61) | 5.40(4.20,7.72) | 17.09(15.68,18.64) |
| *PSFC (hPa)* | | | | | |
| SCC | 1.00(1.00,1.00) | 1.00(1.00,1.00) | 1.00(1.00,1.00) | 0.99(0.99,0.99) | 0.99(0.99,0.99) |
| ME | -0.39(-0.54,-0.16) | -0.33(-0.65,0.10) | -0.49(-0.64,-0.23) | -0.02(-0.28,0.14) | -1.18(-1.41,-0.98) |

[Figure]

**Fig. S5** The biases of the simulated annual mean T2MEAN and PREC in Central Asia during the reference period (1986-2005) relative to the observations.

[Figure]

**Fig. S6** Same as **Fig. S5,** but for the annual mean RH2MEAN and WS10MEAN.

[Figure]

**Fig. S7** Same as **Fig. S5**, but for the annual mean SWD, LWD, and PSFC.

[Figure]

**Fig. S8** Spatial correlation coefficients (SCCs) and root mean square errors (RMSEs) of the simulated annual (ANN), summer (JJA: June-July-August), and winter (DJF: December-January-February) mean T2MEAN over CA with and without adjusting based on the elevation differences between the observations and the model grids. The metrics are calculated based on 58 stations' data across CA.

(3) Please clarify the origin of the station data during evaluation.

Reply: The stations' data is from Global Historical Climatology Network (GHCN) of NOAA National Climatic Data Center.

"Here, we use the records of T2MEAN on 58 stations across CA (see the stars in Fig. 1a) as observations, which as well as the records of PREC on 52 stations (which is used in sect. 4.2, see the circles in Fig. 1a) are from Global Historical Climatology Network (GHCN) of NOAA National Climatic Data Center and have been quality controlled (Qiu et al., 2021)." (L255-259)

**RC3: 'Comment on essd-2021-361', Anonymous Referee #3, 07 Jan 2022**

It is my pleasure to review the paper entitled "HCPD-CA High-resolution climate projection dataset in Central Asia for ecological and hydrological applications" by Qiu et al. The authors produced a high-resolution (9km) climate projection dataset over Central Asia based on the dynamically downscaled results being combined with multiple bias-corrected global climate models. This dataset is assumed to serve as a scientific basis for assessing the impacts of climate change over Central Asia on many sectors.

Given the importance of this work and its potential impact, I would like to providing the following comments for improving the manuscript.

(1) The time range in 2.2 (1981-2005/2026-2050) is different from that in 2.3 (1985-2005/2060-2050). Could you please explain or make them consistent for readable?

Reply: The bias-correction technique used in this study is developed by Bruyère et al. (2014). They produced the bias-corrected CCSM4 outputs (DOI: https://doi.org/10.5065/D6DJ5CN4) with a 25-year base period (1981-2005) during the bias correction. In this study, we produced the bias-corrected MPI-ESM-MR and HadGEM2-ES outputs with the same base period as them. The base period used during the bias correction is not necessary to be consistent with the reference period (1986-2005) of the RCM simulations.

"The bias-corrected CCSM4 outputs (DOI: https://doi.org/10.5065/D6DJ5CN4) is produced by Bruyère et al. (2014) with a 25-year base period (1981-2005) during the bias correction. In this study, we produced the bias-corrected MPI-ESM-MR and HadGEM2-ES outputs with the same base period as them. Note that the base period used during the bias correction is not necessary to be consistent with the reference period (1986-2005) of the RCM simulations." (L148-153)

Ref: Bruyère, C. L., Done, J. M., Holland, G. J., and Fredrick, S.: Bias corrections of global models for regional climate simulations of high-impact weather, Climate Dynamics, 43, 1847-1856, 10.1007/s00382-013-2011-6, 2014.

(2) Fig. 2 should be explained in detail, especially for the right side of the figure. There are some sub-questions are occurred from this figure:

- How to achieve the action named "WRF with the optional combination of physical schemes" with the input from Bias-corrected GCMs and Observation data?
- The term "WRF with the optional combination of physical schemes" seems to run WRF model again, however, WRF model has been executed in previous step. This term can be renamed properly.
- Is the spatial resolution of Bias-corrected GCMs same to the one of the WRF model output? If not, how to match them?

Reply: I described Fig. 2 in the revised manuscript.

"Fig. 2 shows the flow chart to produce the HCPD-CA dataset. The procedure can be divided into four steps. First, multiple-source observational data is used to evaluate the WRF model with different combinations of physical schemes and then we found the optimal combination of physical schemes for the WRF model. Second, the original GCMs are bias corrected and the bias-corrected GCMs are used to drive the WRF model with the optimal combination of physical schemes. Third, we conducted the dynamical downscaling and produced 9-km resolution downscaled results. At last, the HCPD-CA dataset with certain variables and standard file formats is derived from the downscaled results." (L163-170) As the driving data, the bias-corrected GCMs have much coarser spatial resolution than the WRF model outputs.

(6) In Fig.3, the observational data for temperature is CRU data, the ones for precipitation and other variables are ERA data. Could you please give more reasonable explanation?

Reply: We found the CRU TS v4 dataset generally has good performance to describe the climatology of surface air temperature over CA (Qiu et al., 2021). Thus, we used it to evaluate the simulated T2MEAN/T2MAX/T2MIN. Because the rain-gauge-observation merged in the CRU TS v4 dataset is sparse and unevenly distributed over CA, the precipitation data in it has limitations in depicting the climatology of precipitation in CA, especially over the mountainous areas. In addition, the CRU TS v4 dataset does not have other variables (e.g., relative humidity, wind, and shortwave and longwave radiation). As a results, we used the ERA5-Land dataset to evaluate precipitation and other variables.

*Ref: Qiu Y., Feng, J., Yan, Z., Wang, J., and Li, Z.: High-resolution dynamical downscaling for regional climate projection in Central Asia based on bias-corrected multiple GCMs, Climate Dynamics, 10.1007/s00382-021-05934-2, 2021.*

(7) It is recommended to do more comparisons between the simulation from control experiment and the one being combing Bias-correction GCMs, and to show the improvement by introducing the Bias-correction GCMs.

Reply: "In a recent study (Qiu et al., 2021), we conducted the sensitivity experiments of using the bias-correction technique, to quantify its contribution to improving the RCM simulation. The results show that using the bias-correction technique largely reduced the biases in the simulated annual and seasonal precipitation over CA respect to not using it and slightly improved the model's skill in simulating the spatial pattern of precipitation (see Fig. 4 in Qiu et al., 2021)." (L141-147)

*Ref: Qiu Y., Feng, J., Yan, Z., Wang, J., and Li, Z.: High-resolution dynamical downscaling for regional climate projection in Central Asia based on bias-corrected multiple GCMs, Climate Dynamics, 10.1007/s00382-021-05934-2, 2021.*

(8) Please use the data doi linkage instead of the uuid linkage for accessing the dataset in Lines 22-23 and Lines 187-188.

Reply: Revised.

"It has the DOI https://doi.org/10.11888/Meteoro.tpdc.271759 (Qiu, 2021)." (L26)

"The HCPD-CA has the DOI https://doi.org/10.11888/Meteoro.tpdc.271759 (Qiu, 2021)." (L309)

**RC4: 'Comment on essd-2021-361', Anonymous Referee #4, 10 Jan 2022**

This manuscript describes a high-resolution climate projection dataset in central Asia, which could be potentially useful for hydrological or ecological applications. More proofs of its reliability and confidence levels are necessary to present before it could be used in the consequent applications.

(1) The horizontal resolution of 9-km was used in this study. My question is why 9-km? 9-km is an awaked resolution not large as precedent quarter degree, not small as convection-permitting modeling. At this resolution, whether the cumulus parametrization is used or not is still an open question.

Reply: Here, we carried out a study that involves the dynamical downscaling of multiple bias-corrected GCMs for the CA region with an unprecedented horizontal resolution of 9km. The 9-km resolution is much higher than those ($\geq$ 30km) of the previous regional climate projections in CA. With the limitation of the computational and time cost, we did not use a higher resolution (e.g., 3-5km) that allows convection-permitting modeling.

(2) The bias-corrected GCMs were used as forcing in this study. What does the simulation performance look like if the bias correction was not used? The previous study also claimed that similar performances were obtained in the Tibetan Plateau using either reanalysis or GCMs as forcing in the historical period. Major differences only occur in the temporal changes or linear trends. Almost the same historical simulations were also presented in figures 3-6 using three different forcings GCMs in this study. Therefore, it would be good to present the differences between using bias-corrected GCM or not.

Reply: "In a recent study (Qiu et al., 2021), we conducted the sensitivity experiments of using the bias-correction technique, to quantify its contribution to improving the RCM simulation. The results show that using the bias-correction technique largely reduced the biases in the simulated annual and seasonal precipitation over CA respect to not using it and slightly improved the model's skill in simulating the spatial pattern of precipitation (see Fig. 4 in Qiu et al., 2021)." (L141-147)

*Ref: Qiu Y., Feng, J., Yan, Z., Wang, J., and Li, Z.: High-resolution dynamical downscaling for regional climate projection in Central Asia based on bias-corrected multiple GCMs, Climate Dynamics, 10.1007/s00382-021-05934-2, 2021.*

(3) The future changes in 2031-2050 compared to 1986-2005 are studied. It is neither the end of this century 2100 nor the target year of zero carbon emission 2060. It would be better to describe the importance of this period in the future in central Asia. In addition, all the future changes are based on the model. Suggest adding historical changes or linear trends to solid the credibility or reliability of future changes.

Reply: "As reported in the 1.5°C special report of the Intergovernmental Panel on Climate Chane (IPCC), we are on track to exceed 1.5°C warming between 2030 and 2052 based on the current warming rate, and hence the near-term future projection becomes more critical to human development than that for the end of this century. Therefore, this study focuses on projected climate changes over CA in the near-term future (2031-2050). Long-term continuous (e.g., 1986-2100) regional climate projections in CA are more useful for studies in this region and will be conducted in the next stage." (L290-296) We have discussed our findings with the previous studies in a recently published paper (Qiu et al., 2021). For instance, enhanced warming projected in many mountains in the world is not found in CA, which is consistent with the study based on the reanalysis datasets during the past (Hu et al, 2014). Stronger warming is detected in the north part of Central Asia, consistent with the previous regional climate projections (Mannig et al., 2013;Ozturk et al., 2017;Peng et al., 2019). A comprehensive comparison between the future and historical changes based on multiple methods and multi-source data by our group is in process.

*Ref:*

*Hu, Z., Zhang, C., Hu, Q., and Tian, H.: Temperature Changes in Central Asia from 1979 to 2011 Based on Multiple Datasets, 27, 1143-1167, 10.1175/jcli-d-13-00064.1, 2014.*
*Mannig, B., Müller, M., Starke, E., Merkenschlager, C., Mao, W., Zhi, X., Podzun, R., Jacob, D., and Paeth,*

H.: Dynamical downscaling of climate change in Central Asia, Global and Planetary Change, 110, 26-39, https://doi.org/10.1016/j.gloplacha.2013.05.008, 2013.

Ozturk, T., Turp, M. T., Türkeş, M., and Kurnaz, M. L.: Projected changes in temperature and precipitation climatology of Central Asia CORDEX Region 8 by using RegCM4.3.5, Atmospheric Research, 183, 296-307, https://doi.org/10.1016/j.atmosres.2016.09.008, 2017.

Peng, D., Zhou, T., Zhang, L., and Zou, L. J. C. D.: Detecting human influence on the temperature changes in Central Asia, 53, 4553-4568, 2019.

Qiu, Y., Feng, J., Yan, Z., Wang, J., and Li, Z.: High-resolution dynamical downscaling for regional climate projection in Central Asia based on bias-corrected multiple GCMs, Climate Dynamics, 10.1007/s00382-021-05934-2, 2021.